# SCORE REGULARIZED POLICY OPTIMIZATION THROUGH DIFFUSION BEHAVIOR

**Huayu Chen[1], Cheng Lu[1], Zhengyi Wang[1], Hang Su[1,2], Jun Zhu[1,2]***

[1]Department of Computer Science & Technology, Institute for AI, BNRist Center,
Tsinghua-Bosch Joint ML Center, THBI Lab, Tsinghua University
[2]Pazhou Laboratory (Huangpu), Guangzhou, Guangdong
{chenhuay21,wang-zy21}@mails.tsinghua.edu.cn;
lucheng.lc15@gmail.com; {suhangss,dcszj}@tsinghua.edu.cn

## ABSTRACT

Recent developments in offline reinforcement learning have uncovered the immense potential of diffusion modeling, which excels at representing heterogeneous behavior policies. However, sampling from diffusion policies is considerably slow because it necessitates tens to hundreds of iterative inference steps for one action. To address this issue, we propose to extract an efficient deterministic inference policy from critic models and pretrained diffusion behavior models, leveraging the latter to directly regularize the policy gradient with the behavior distribution's score function during optimization. Our method enjoys powerful generative capabilities of diffusion modeling while completely circumventing the computationally intensive and time-consuming diffusion sampling scheme, both during training and evaluation. Extensive results on D4RL tasks show that our method boosts action sampling speed by more than 25 times compared with various leading diffusion-based methods in locomotion tasks, while still maintaining state-of-the-art performance. Code: https://github.com/thu-ml/SRPO.

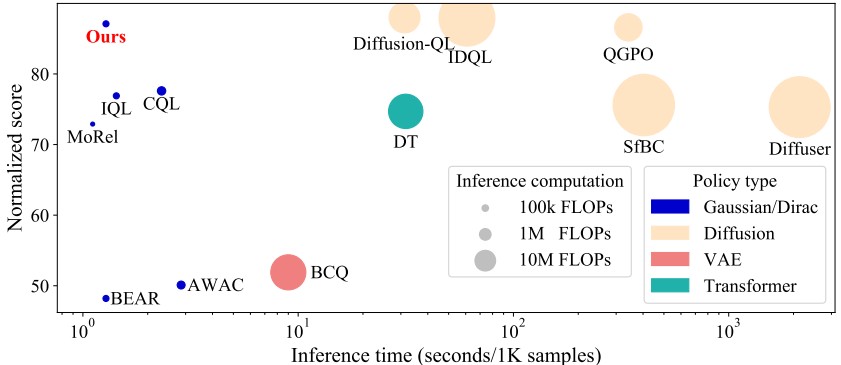

Figure 1: Performance and computational efficiency of different algorithms in D4RL Locomotion tasks. Computation time is assessed using a consistent hardware setup and PyTorch backend.

## 1 INTRODUCTION

Offline reinforcement learning (RL) aims to tackle decision-making problems by solely utilizing a pre-collected behavior dataset. This offers a practical solution for tasks where data collection can be associated with substantial risks or exorbitant costs. A central challenge for offline RL is the realization of behavior regularization, which entails ensuring the learned policy stays in support of the behavior distribution. Weighted regression (Peng et al., 2019; Kostrikov et al., 2022) provides a promising approach that directly utilizes behavioral actions as sources of supervision for policy training. Another prevalent approach is behavior-regularized policy optimization (Kumar et al., 2019;

---

*Corresponding author.

Wu et al., 2019), which builds a generative behavior model, followed by constraining the divergence between the learned and the behavior model during policy optimization.

An expressive generative model holds a pivotal role in the aforementioned regularization methods. In weighted regression, a unimodal actor is prone to suffer from the "mode covering" issue, a phenomenon where policies end up selecting out-of-support actions in the middle region between two behavioral modes (Wang et al., 2023a; Hansen-Estruch et al., 2023). Expressive policy classes, with diffusion models (Ho et al., 2020) as prime choices, help to resolve this issue. In behavior-regularized policy optimization, diffusion modeling can also be significantly advantageous for an accurate estimate of policy divergence, due to its strong ability to represent heterogeneous behavior datasets, outperforming conventional methods like Gaussians or variational auto-encoders (VAEs) (Goo & Niekum, 2022; Chen et al., 2023).

However, a major drawback of utilizing diffusion models in offline RL is the considerably slow sampling speed – diffusion policies usually require 5-100 iterative inference steps to create an action sample. Moreover, diffusion policies tend to be excessively stochastic, forcing the generation of dozens of action candidates in parallel to pinpoint the final optimal one (Wang et al., 2023a). As existing methods necessitate sampling from or backpropagating through diffusion policies during training and evaluation, it has significantly slowed down experimentation and limited the application in fields that are computationally sensitive or require high control frequency, such as robotics. Therefore, it is critical to systematically investigate the question: *is it feasible to fully exploit the generative capabilities of diffusion models without directly sampling actions from them?*

In this paper, we propose **S**core **R**egularized **P**olicy **O**ptimization (SRPO) with a positive answer to the above question. The basic idea is to extract a simple deterministic inference policy from critic and diffusion behavior models to avoid the iterative diffusion sampling process during evaluation. To achieve this, we show that the gradient of the divergence term in regularized policy optimization is essentially related to the score function of behavior distribution. The latter can be effectively approximated by any pretrained score-based model including diffusion models (Song et al., 2021). This allows us to directly regularize the policy *gradient* instead of the policy *loss*, removing the need to generate fake behavioral actions for policy-divergence estimation (Section 3).

We develop a practical algorithm to solve continuous control tasks (Section 4) by combining SRPO with implicit Q-learning (Kostrikov et al., 2022) and continuous-time diffusion behavior modeling (Lu et al., 2023). For policy extraction, we incorporate similar techniques that have facilitated recent advances in text-to-3D research such as DreamFusion (Poole et al., 2023). These include leveraging an ensemble of score approximations under different diffusion times to exploit the pretrained behavior model and a baseline term to reduce variance for gradient estimation. We empirically show that these techniques successfully help improve performance and stabilize training for policy extraction.

We evaluate our method in D4RL tasks (Fu et al., 2020). Results demonstrate that our method enjoys a more than $25\times$ boost in action sampling speed and less than $1\%$ of computational cost for evaluation compared with several leading diffusion-based methods while maintaining similar overall performance in locomotion tasks (Figure 1). We also conduct 2D experiments to better illustrate that SRPO successfully constrains the learned policy close to various complex behavior distributions.

## 2 BACKGROUND

### 2.1 OFFLINE REINFORCEMENT LEARNING

Consider a typical Markov Decision Process (MDP) described by the tuple $\langle \mathcal{S}, \mathcal{A}, P, r, \gamma \rangle$, where $\mathcal{S}$ is the state space, $\mathcal{A}$ the action space, $P(\boldsymbol{s}'|\boldsymbol{s}, \boldsymbol{a})$ the transition function, $r(\boldsymbol{s}, \boldsymbol{a})$ the reward function and $\gamma$ the discount factor. The goal of reinforcement learning (RL) is to train a parameterized policy $\pi_\theta(\boldsymbol{a}|\boldsymbol{s})$ which maximizes the expected episode return. Offline RL relies solely on a static dataset $\mathcal{D}^\mu$ containing interacting history $\{\boldsymbol{s}, \boldsymbol{a}, r, \boldsymbol{s}'\}$ between a behavior policy $\mu(\boldsymbol{a}|\boldsymbol{s})$ and the environment to train the parameterized policy.

Suppose we can evaluate the quality of a given action by estimating its expected return-to-go using a Q-network $Q_\phi(\boldsymbol{s}, \boldsymbol{a}) \approx Q^\pi(\boldsymbol{s}, \boldsymbol{a}) := \mathbb{E}_{\boldsymbol{s}_1=\boldsymbol{s}, \boldsymbol{a}_1=\boldsymbol{a}; \pi}[\sum_{n=1}^{\infty} \gamma^n r(\boldsymbol{s}_n, \boldsymbol{a}_n)]$, we can formulate the training objective of offline RL as $\max_\pi \mathbb{E}_{\boldsymbol{s}\sim\mathcal{D}^\mu, \boldsymbol{a}\sim\pi(\cdot|\boldsymbol{s})} Q_\phi(\boldsymbol{s}, \boldsymbol{a}) - \frac{1}{\beta} D_{\mathrm{KL}}\left[\pi(\cdot|\boldsymbol{s})||\mu(\cdot|\boldsymbol{s})\right]$ (Wu et al., 2019). Note that a KL regularization term is added mainly to ensure the learned policy stays in

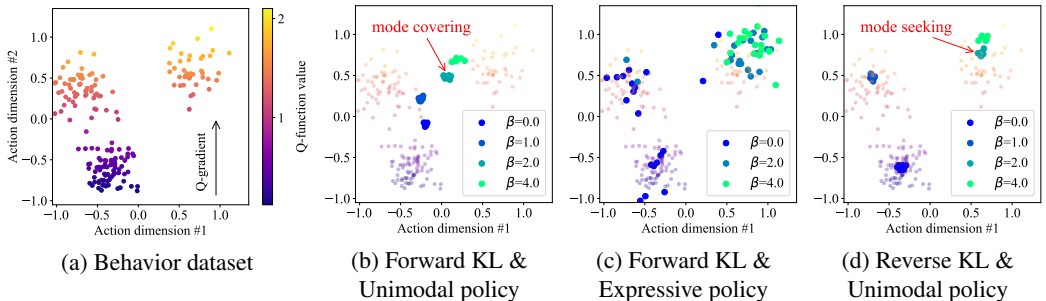

Figure 2: Comparison of different policy extraction methods under bandit settings. Forward KL policy extraction is prone to generate out-of-support actions if the policy is not sufficiently expressive (e.g., Gaussians). This can be mitigated either by employing a more expressive policy class or by switching to a reverse KL objective (our method), which demonstrates a mode-seeking nature.

support of explored actions. $\beta$ is some temperature coefficient. Previous work (Peters et al., 2010; Peng et al., 2019) has shown that the optimal policy for such an optimization problem is

$$\pi^*(\boldsymbol{a}|\boldsymbol{s}) = \frac{1}{Z(\boldsymbol{s})}\,\mu(\boldsymbol{a}|\boldsymbol{s})\exp\left(\beta Q_\phi(\boldsymbol{s},\boldsymbol{a})\right), \tag{1}$$

where $Z(\boldsymbol{s})$ is the partition function. The core problem for offline RL now becomes how to efficiently model and sample from the optimal policy distribution $\pi^*(\cdot|\boldsymbol{s})$.

## 2.2 OPTIMAL POLICY EXTRACTION

Existing methods to explicitly model $\pi^*$ with a parameterized policy $\pi_\theta$ can be roughly divided into two main categories—*weighted regression* and *behavior-regularized policy optimization*:

$$\min_\theta \mathbb{E}_{\boldsymbol{s}\sim\mathcal{D}^\mu} \underbrace{D_{\mathrm{KL}}\left[\pi^*(\cdot|\boldsymbol{s})||\pi_\theta(\cdot|\boldsymbol{s})\right]}_{Forward\ \mathrm{KL}} \Leftrightarrow \max_\theta \mathbb{E}_{(\boldsymbol{s},\boldsymbol{a})\sim\mathcal{D}^\mu}\underbrace{\left[\frac{1}{Z(\boldsymbol{s})}\log\pi_\theta(\boldsymbol{a}|\boldsymbol{s})\,e^{\beta Q_\phi(\boldsymbol{s},\boldsymbol{a})}\right]}_{Weighted\ Regression}, \tag{2}$$

$$\updownarrow$$

$$\min_\theta \mathbb{E}_{\boldsymbol{s}\sim\mathcal{D}^\mu}\underbrace{D_{\mathrm{KL}}\left[\pi_\theta(\cdot|\boldsymbol{s})||\pi^*(\cdot|\boldsymbol{s})\right]}_{Reverse\ \mathrm{KL}} \Leftrightarrow \max_\theta \underbrace{\mathbb{E}_{\boldsymbol{s}\sim\mathcal{D}^\mu,\boldsymbol{a}\sim\pi_\theta}Q_\phi(\boldsymbol{s},\boldsymbol{a}) - \frac{1}{\beta}D_{\mathrm{KL}}\left[\pi_\theta(\cdot|\boldsymbol{s})||\mu(\cdot|\boldsymbol{s})\right]}_{Behavior\text{-}Regularized\ Policy\ Optimization}. \tag{3}$$

Weighted regression directly utilizes behavioral actions as sources of supervision for policy training. This circumvents the necessity to explicitly model the intricate behavior policy but leads to another mode-covering issue due to the objective's forward-KL nature (Eq. (2)). The direct consequence is that weighted regression algorithms display sensitivity to the proportion of suboptimal data in the dataset (Yue et al., 2022), especially when the policy model lacks distributional expressivity (Chen et al., 2023), as is depicted in Figure 2. Recent work (Wang et al., 2023a; Lu et al., 2023) attempts to alleviate this by employing more expressive policy classes, such as diffusion models (Ho et al., 2020). However, these methods usually compromise on computational efficiency (Kang et al., 2023).

In comparison, behavior-regularized policy optimization (Wu et al., 2019) emerges as a more suitable approach for training simpler policy models such as Gaussian models. This fundamentally stems from its basis on a reverse-KL objective (Eq. (3)), which inherently encourages a mode-seeking behavior. However, approximating the second KL term in Eq. (3) is usually difficult. Regarding this, in practical implementation previous studies (Kumar et al., 2019; Wu et al., 2019; Xu et al., 2021) usually first construct generative behavior models to approximate the policy-behavior divergence.

## 2.3 DIFFUSION MODELS FOR SCORE FUNCTION ESTIMATION

Diffusion models (Sohl-Dickstein et al., 2015; Ho et al., 2020; Song et al., 2021) are powerful generative models. They operate by defining a forward diffusion process to perturb the data distribution into a noise distribution for training the diffusion model. Subsequently, this model is employed to reverse the diffusion process, thereby generating data samples from pure noise.

In particular, the forward process is conducted by gradually adding Gaussian noise to samples $\boldsymbol{x}_0$ from an unknown data distribution $q_0(\boldsymbol{x}_0) := q(\boldsymbol{x})$ at time 0, forming a series of diffused distributions $q_t(\boldsymbol{x}_t)$ at time $t$. The transition distribution $q_{t0}(\boldsymbol{x}_t|\boldsymbol{x}_0)$ is:

$$q_{t0}(\boldsymbol{x}_t|\boldsymbol{x}_0) = \mathcal{N}(\boldsymbol{x}_t|\alpha_t\boldsymbol{x}_0, \sigma_t^2\boldsymbol{I}), \qquad \text{which implies} \qquad \boldsymbol{x}_t = \alpha_t\boldsymbol{x}_0 + \sigma_t\boldsymbol{\epsilon}. \tag{4}$$

Here, $\alpha_t, \sigma_t > 0$ are manually defined, and $\boldsymbol{\epsilon}$ is random Gaussian noise.

For the reverse process, Ho et al. (2020) train a diffusion model $\boldsymbol{\epsilon}_\theta(\boldsymbol{x}_t|t)$ to predict the noise added to the diffused sample $\boldsymbol{x}_t$ in order to iteratively reconstruct $\boldsymbol{x}_0$. The optimization problem is

$$\min_\theta \mathbb{E}_{t,\boldsymbol{x}_0,\boldsymbol{\epsilon}} \left[\|\boldsymbol{\epsilon}_\theta(\boldsymbol{x}_t|t) - \boldsymbol{\epsilon}\|_2^2\right]. \tag{5}$$

More formally, Song et al. (2021) show that diffusion models are in essence estimating the *score function* $\nabla_{\boldsymbol{x}_t} \log q_t(\boldsymbol{x}_t)$ of the diffused data distribution $q_t$, such that:

$$\nabla_{\boldsymbol{x}_t} \log q_t(\boldsymbol{x}_t) = -\boldsymbol{\epsilon}^*(\boldsymbol{x}_t|t)/\sigma_t \approx -\boldsymbol{\epsilon}_\theta(\boldsymbol{x}_t|t)/\sigma_t, \tag{6}$$

and the reverse diffusion process can alternatively be interpreted as discretizing an ODE:

$$\frac{\mathrm{d}\boldsymbol{x}_t}{\mathrm{d}t} = f(t)\boldsymbol{x}_t - \frac{1}{2}g^2(t)\nabla_{\boldsymbol{x}_t} \log q_t(\boldsymbol{x}_t), \tag{7}$$

where $f(t) = \frac{\mathrm{d}\log\alpha_t}{\mathrm{d}t}, g^2(t) = \frac{\mathrm{d}\sigma_t^2}{\mathrm{d}t} - 2\frac{\mathrm{d}\log\alpha_t}{\mathrm{d}t}\sigma_t^2$, leaving $\nabla_{\boldsymbol{x}_t} \log q_t(\boldsymbol{x}_t)$ as the only unknown term. In offline RL, diffusion models have been discovered as an effective tool for modeling heterogeneous behavior policies. Usually states $\boldsymbol{s}$ are considered as conditions while actions $\boldsymbol{a}$ are considered as data points $\boldsymbol{x}$, such that a conditional diffusion model $\boldsymbol{\epsilon}(\boldsymbol{a}_t|\boldsymbol{s}, t)$ can be constructed to represent $\mu(\boldsymbol{a}|\boldsymbol{s})$.

## 3 SCORE REGULARIZED POLICY OPTIMIZATION

In this paper, we seek to learn a deterministic policy $\pi_\theta$ to capture the mode of a potentially complex policy distribution $\pi^*$ introduced in Eq. (1). To achieve this, we employ a reverse-KL policy extraction scheme (Eq. (3)) given its mode-seeking nature:

$$\max \mathcal{L}_\pi(\theta) = \mathbb{E}_{\boldsymbol{s}\sim\mathcal{D}^\mu,\boldsymbol{a}\sim\pi_\theta} Q_\phi(\boldsymbol{s},\boldsymbol{a}) - \frac{1}{\beta}D_{\mathrm{KL}}\left[\pi_\theta(\cdot|\boldsymbol{s})||\mu(\cdot|\boldsymbol{s})\right]. \tag{8}$$

Solving the above optimization problem requires estimating $D_{\mathrm{KL}}\left[\pi_\theta(\cdot|\boldsymbol{s})||\mu(\cdot|\boldsymbol{s})\right]$. Regarding this, previous research (Kumar et al., 2019; Wu et al., 2019; Xu et al., 2021; Wu et al., 2022) use sample-based methods: first constructing a behavioral model $\mu_\psi \approx \mu$, followed by sampling fake actions from $\mu_\psi(\cdot|\boldsymbol{s})$ and $\pi_\theta(\cdot|\boldsymbol{s})$ to approximate the policy divergence. However, this approach necessitates sampling from the behavior model during training, imposing a substantial computational burden. This drawback is exacerbated when employing expressive yet sampling-expensive behavior models such as diffusion models.

We propose an alternative way to solve Eq. (8). By decomposing the KL term, we can get

$$\mathcal{L}_\pi(\theta) = \underbrace{\mathbb{E}_{\boldsymbol{s}\sim\mathcal{D}^\mu,\boldsymbol{a}\sim\pi_\theta} Q_\phi(\boldsymbol{s},\boldsymbol{a})}_{\text{Policy optimization}} + \frac{1}{\beta}\underbrace{\mathbb{E}_{\boldsymbol{s}\sim\mathcal{D}^\mu,\boldsymbol{a}\sim\pi_\theta}\log\mu(\boldsymbol{a}|\boldsymbol{s})}_{\text{Behavior regularization}} + \frac{1}{\beta}\underbrace{\mathbb{E}_{\boldsymbol{s}\sim\mathcal{D}^\mu}\mathcal{H}(\pi_\theta(\cdot|\boldsymbol{s}))}_{\text{Entropy (often constant}^1)}. \tag{9}$$

Then we calculate the gradient of Eq. (9) under the condition that $\pi_\theta$ is deterministic. Applying the chain rule and the reparameterization trick, we have:

$$\nabla_\theta\mathcal{L}_\pi(\theta) = \mathbb{E}_{\boldsymbol{s}\sim\mathcal{D}^\mu}\left[\nabla_{\boldsymbol{a}}Q_\phi(\boldsymbol{s},\boldsymbol{a})|_{\boldsymbol{a}=\pi_\theta(\boldsymbol{s})} + \frac{1}{\beta}\underbrace{\nabla_{\boldsymbol{a}}\log\mu(\boldsymbol{a}|\boldsymbol{s})|_{\boldsymbol{a}=\pi_\theta(\boldsymbol{s})}}_{=-\boldsymbol{\epsilon}^*(\boldsymbol{a}_t|\boldsymbol{s},t)/\sigma_t|_{t\to 0} \;\;(\text{by Eq. 6})}\right]\nabla_\theta\pi_\theta(\boldsymbol{s}). \tag{10}$$

It is noted that the only unknown term above is the score function $\nabla_{\boldsymbol{a}}\log\mu(\boldsymbol{a}|\boldsymbol{s})$ of the behavior distribution. Our key insight is that a pretrained diffusion behavior model can be leveraged to

---

[1]In this paper we only consider the cases where $\pi_\theta$ is an isotropic Gaussian with fixed variance. For brevity, we informally view Dirac as Gaussian whose variance is infinitesimally small.

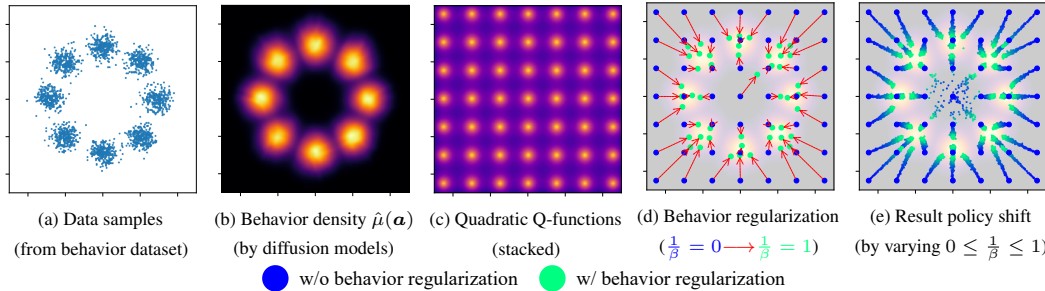

(a) Data samples     (b) Behavior density $\hat{\mu}(\boldsymbol{a})$   (c) Quadratic Q-functions   (d) Behavior regularization   (e) Result policy shift

(from behavior dataset)     (by diffusion models)     (stacked)    ($\frac{1}{\beta} = 0 \longrightarrow \frac{1}{\beta} = 1$)   (by varying $0 \le \frac{1}{\beta} \le 1$)

● w/o behavior regularization   ● w/ behavior regularization

Figure 3: Illustration of SRPO in 2D bandit settings. **(a)** A predefined complex data distribution, which represents the potentially heterogeneous behavior policy $\mu(\boldsymbol{a})$. **(b)** A diffusion model $\hat{\mu}(\boldsymbol{a})$ is trained to fit the behavior distribution. The data density can be analytically calculated based on Song et al. (2021). **(c)** The Q-function is manually defined as a quadratic function: $Q(\boldsymbol{a}) := -(\boldsymbol{a} - \boldsymbol{a}_{\text{tar}})^2$, where $\boldsymbol{a}_{\text{tar}}$ represents the 2D point with the highest estimated Q-value and is selected from a set of grid intersections. These individual Q-functions with different $\boldsymbol{a}_{\text{tar}}$ are depicted together in a stacked way in Figure (c). **(d)&(e)** By optimizing deterministic policies $\pi(\cdot) = \boldsymbol{a}_{\text{reg}}$ according to Eq. (10) and tuning the temperature coefficient $\beta$, resulting policies shift from greedy ones which tend to maximize corresponding Q-functions to conservative ones which are successfully constrained close to the behavior distribution. See more experimental results in Appendix A.

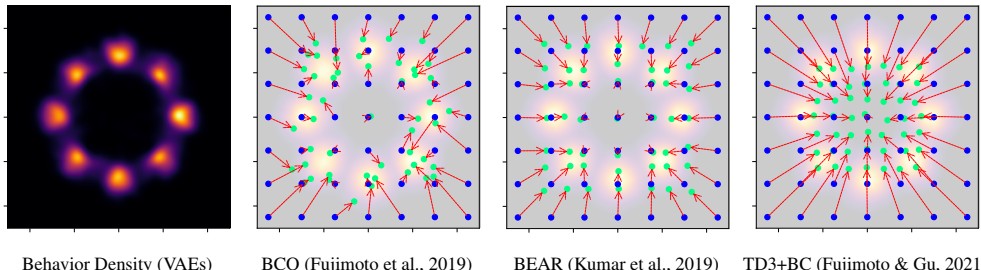

Behavior Density (VAEs)    BCQ (Fujimoto et al., 2019)    BEAR (Kumar et al., 2019)    TD3+BC (Fujimoto & Gu, 2021)

Figure 4: Performance of other behavior regularization methods. See more results in Appendix A.

effectively estimate this term. This is because diffusion models $\boldsymbol{\epsilon}(\boldsymbol{x}|t)$ are essentially approximating the score function $\nabla_{\boldsymbol{x}} \log \mu_t(\boldsymbol{x})$ of the diffused data distribution $\mu_t(\boldsymbol{x}_t)$ (Eq. (6)).

Specifically, we first pretrain a diffusion behavior model, denoted as $\boldsymbol{\epsilon}(\boldsymbol{a}_t|\boldsymbol{s}, t)$, to approximate $\nabla_{\boldsymbol{a}} \log \mu(\boldsymbol{a}|\boldsymbol{s})$. By doing so, we can regularize the optimization process of another deterministic actor $\pi_\theta$. We term our method as **S**core **R**egularized **P**olicy **O**ptimization (SRPO), given its distinctive feature of performing regularization at the gradient level, as opposed to the loss function level. Figure 3 provides a 2D bandit example of SRPO.

Compared with previous work (Wang et al., 2023a; Lu et al., 2023; Hansen-Estruch et al., 2023) that directly trains a diffusion policy for inference in evaluation, the main advantage of SRPO is its computational efficiency. SRPO entirely circumvents the computationally demanding action sampling scheme associated with the diffusion process. Yet, it still taps into the robust generative strengths of diffusion models, especially their ability to represent potentially diverse behavior datasets.

## 4 PRACTICAL ALGORITHM

In this section, we derive a practical algorithm for applying SRPO in offline RL (Algorithm 1). The algorithm includes three parts: implicit Q-learning (Section 4.1); diffusion-based behavior modeling (Section 4.1); and score-regularized policy extraction (Section 4.2).

### 4.1 PRETRAINING THE DIFFUSION BEHAVIOR MODEL AND Q-NETWORKS

For Q-networks, we choose to use implicit Q-learning (Kostrikov et al., 2022) to decouple critic training from actor training. The core ingredient of the training pipeline is expectile regression, which

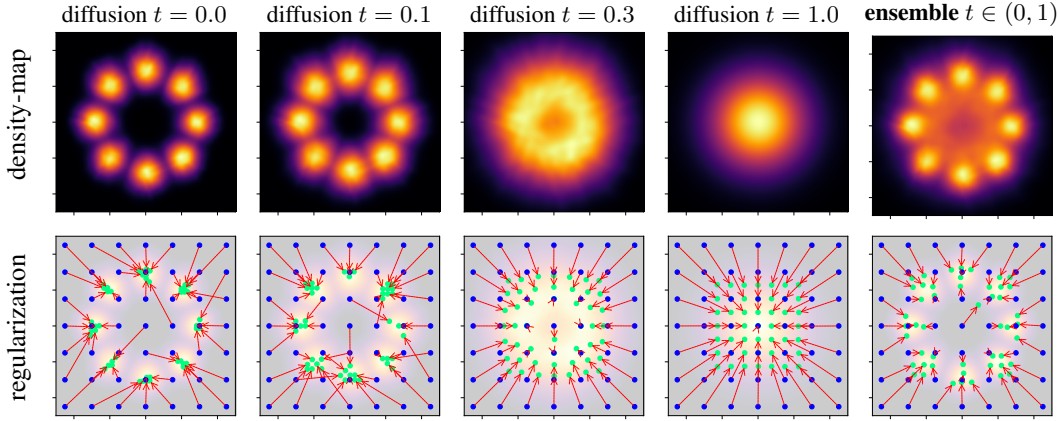

Figure 5: Empirical benefits of ensembling multiple diffusion times. See Remark 1 in Appendix B for a detailed explanation.

requires only sampling actions from existing datasets for bootstrapping:

$$\min_\zeta \mathcal{L}_V(\zeta) = \mathbb{E}_{(\boldsymbol{s},\boldsymbol{a})\sim\mathcal{D}^\mu}[L_2^\tau(Q_\phi(\boldsymbol{s},\boldsymbol{a}) - V_\zeta(\boldsymbol{s}))], \text{ where } L_2^\tau(\boldsymbol{u}) = |\tau - \mathbb{1}(\boldsymbol{u} < 0)|\boldsymbol{u}^2, \quad (11)$$

$$\min_\phi \mathcal{L}_Q(\phi) = \mathbb{E}_{(\boldsymbol{s},\boldsymbol{a},\boldsymbol{s}')\sim\mathcal{D}^\mu}\left[\|r(\boldsymbol{s},\boldsymbol{a}) + \gamma V_\zeta(\boldsymbol{s}') - Q_\phi(\boldsymbol{s},\boldsymbol{a})\|_2^2\right]. \quad (12)$$

When the expectile parameter $\tau \in (0,1)$ is larger than 0.5, the asymmetric L2-objective $\mathcal{L}_V(\zeta)$ would downweight suboptimal actions which have lower Q-values, removing the need of an explicit policy.

For behavior models, in order to represent the behavior distribution with high fidelity and estimate its score function, we follow previous work (Chen et al., 2023; Hansen-Estruch et al., 2023) and train a conditional behavior cloning model:

$$\min_\psi \mathcal{L}_\mu(\psi) = \mathbb{E}_{t,\boldsymbol{\epsilon},(\boldsymbol{s},\boldsymbol{a})\sim\mathcal{D}^\mu}\left[\|\boldsymbol{\epsilon}_\psi(\boldsymbol{a}_t|\boldsymbol{s},t) - \boldsymbol{\epsilon}\|_2^2\right]_{\boldsymbol{a}_t=\alpha_t\boldsymbol{a}+\sigma_t\boldsymbol{\epsilon}}, \quad (13)$$

where $t \sim \mathcal{U}(0,1)$ and $\boldsymbol{\epsilon} \sim \mathcal{N}(\boldsymbol{0},\boldsymbol{I})$. The model architecture of $\boldsymbol{\epsilon}_\psi$ is consistent with the one proposed by Hansen-Estruch et al. (2023), with the sole difference being that our model uses continuous-time inputs similar to Lu et al. (2023) instead of discrete ones as used by Wang et al. (2023a) and Hansen-Estruch et al. (2023).

Once we have finished training the behavior model $\boldsymbol{\epsilon}_\psi$, we can use it to estimate the score function of $\mu_t$, the diffused distribution of $\mu(\boldsymbol{a}|\boldsymbol{s})$ at time $t$. We have $\nabla_{\boldsymbol{a}_t} \log \mu_t(\boldsymbol{a}_t|\boldsymbol{s},t) = -\boldsymbol{\epsilon}_\psi^*(\boldsymbol{a}_t|\boldsymbol{s},t)/\sigma_t \approx -\boldsymbol{\epsilon}_\psi(\boldsymbol{a}_t|\boldsymbol{s},t)/\sigma_t$ as shown by Song et al. (2021).

---

**Algorithm 1** SRPO

Initialize parameters $\psi, \zeta, \phi, \theta$.
*// Critic training (IQL)*
**for** each gradient step **do**
   $\zeta \leftarrow \zeta - \lambda_V \nabla_\zeta L_V(\zeta)$ (Eq. 11)
   $\phi \leftarrow \phi - \lambda_Q \nabla_\phi L_Q(\phi)$ (Eq. 12)
*// Behavior training*
**for** each gradient step **do**
   $\psi \leftarrow \psi - \lambda_\mu \nabla_\psi L_\mu(\psi)$ (Eq. 13)
*// Policy extraction*
**for** each gradient step **do**
   $\theta \leftarrow \theta + \lambda_\pi \nabla_\theta \mathcal{L}_\pi^{\text{surr}}(\theta)$ (Eq. 15)

---

### 4.2 POLICY EXTRACTION FROM PRETRAINED MODELS

The policy extraction scheme proposed in Section 3 only leverages the pretrained diffusion behavior model $\boldsymbol{\epsilon}_\phi(\boldsymbol{a}_t|\boldsymbol{s},t)$ at time $t \to 0$, where the behavior distribution $\mu$ has not been diffused. However, $\boldsymbol{\epsilon}_\phi(\boldsymbol{a}_t|\boldsymbol{s},t)$ is trained to represent a series of diffused behavior distributions $\mu_t$ at various times $t \in (0,1)$. In order to exploit the generative capacity of $\boldsymbol{\epsilon}_\phi$, we replace the original training objective $\mathcal{L}_\pi(\theta)$ with a new surrogate objective:

$$\max_\theta \mathcal{L}_\pi^{\text{surr}}(\theta) = \mathbb{E}_{\boldsymbol{s},\boldsymbol{a}\sim\pi_\theta}Q_\phi(\boldsymbol{s},\boldsymbol{a}) - \frac{1}{\beta}\mathbb{E}_{t,\boldsymbol{s}}\omega(t)\frac{\sigma_t}{\alpha_t}D_{\text{KL}}\left[\pi_{t,\theta}(\cdot|\boldsymbol{s})\|\mu_t(\cdot|\boldsymbol{s})\right], \quad (14)$$

where $t \sim \mathcal{U}(0.02, 0.98)$, $\boldsymbol{s} \sim \mathcal{D}^\mu$. Both $\mu_t$ and $\pi_{\theta,t}$ follow the same forward diffusion process in Eq. (4), where $\mu_t(\boldsymbol{a}_t|\boldsymbol{s}) := \mathbb{E}_{\boldsymbol{a}\sim\mu(\cdot|\boldsymbol{s})}\mathcal{N}(\boldsymbol{a}_t|\alpha_t\boldsymbol{a},\sigma_t^2\boldsymbol{I})$, and $\pi_{\theta,t}(\boldsymbol{a}_t|\boldsymbol{s}) :=$

$\mathbb{E}_{\boldsymbol{a} \sim \pi_\theta(\cdot|\boldsymbol{s})} \mathcal{N}(\boldsymbol{a}_t|\alpha_t \boldsymbol{a}, \sigma_t^2 \boldsymbol{I})$. $\omega(t)$ is a weighting function that adjusts the importance of each time $t$. We can nearly recover $\mathcal{L}_\pi(\theta)$ by setting $\omega(t)$ to $\delta(t-0.02)\frac{\alpha_{0.02}}{\sigma_{0.02}}$ (ablation studies in Section 6.3).

Empirically, the surrogate objective $\mathcal{L}_\pi^{\mathrm{surr}}(\theta)$ ensembles various diffused behavior policies $\mu_t$ to regularize training of the same parameterized policy $\pi_\theta$. This is supported by an observation (**Proposition** 1 in Appendix B): $\arg\min_\pi D_{\mathrm{KL}}[\pi_t(\cdot|\boldsymbol{s})||\mu_t(\cdot|\boldsymbol{s})] = \arg\min_\pi D_{\mathrm{KL}}[\pi(\cdot|\boldsymbol{s})||\mu(\cdot|\boldsymbol{s})]$.

Similarly to Section 3, we can optimize Eq. (14) by calculating its gradient:

**Proposition 2.** *(Proof in Appendix B) Given that $\pi_\theta$ is deterministic ($\boldsymbol{a} = \pi_\theta(\boldsymbol{s})$) such that $\pi_{\theta,t}$ is Gaussian ($\boldsymbol{a}_t = \alpha_t \boldsymbol{a} + \sigma_t \boldsymbol{\epsilon},\ \boldsymbol{\epsilon} \sim \mathcal{N}(\mathbf{0}, \boldsymbol{I})$), the gradient for optimizing $\max \mathcal{L}_\pi^{surr}(\theta)$ satisfies*

$$\nabla_\theta \mathcal{L}_\pi^{surr}(\theta) \approx \left[ \mathbb{E}_{\boldsymbol{s}} \nabla_{\boldsymbol{a}} Q_\phi(\boldsymbol{s}, \boldsymbol{a})|_{\boldsymbol{a}=\pi_\theta(\boldsymbol{s})} - \frac{1}{\beta} \mathbb{E}_{t,\boldsymbol{s},\boldsymbol{\epsilon}} \omega(t) (\boldsymbol{\epsilon}_\psi(\boldsymbol{a}_t|\boldsymbol{s},t) \underbrace{-\boldsymbol{\epsilon}}_{\text{subtracted baseline}})|_{\boldsymbol{a}_t=\alpha_t \pi_\theta(\boldsymbol{s})+\sigma_t \boldsymbol{\epsilon}} \right] \nabla_\theta \pi_\theta(\boldsymbol{s}). \quad (15)$$

Note that we additionally subtract the action noise $\boldsymbol{\epsilon}$ from $\boldsymbol{\epsilon}_\psi(\boldsymbol{a}_t|\boldsymbol{s},t)$ in the above equation. This term does not influence the expected value of $\nabla_\theta \mathcal{L}_\pi^{\mathrm{surr}}(\theta)$ but could reduce the estimation variance since it is correlated with $\boldsymbol{\epsilon}_\psi(\boldsymbol{a}_t|\boldsymbol{s},t)$ (See Appendix B). We refer to it as the *baseline* term because it is similar to the subtracted baseline in Policy Gradient (Sutton & Barto, 1998) algorithms.

The idea of ensembling diffused behavior policies for regularization and subtracting the baseline term to reduce estimation variance both draw inspiration from the latest developments in text-to-3D research such as DreamFusion (Poole et al., 2023). We elaborate more on the connection between the two work in Section 5 and ablate these techniques in the realm of continuous control in Section 6.3.

## 5 RELATED WORK

**Behavior Regularization in Offline Reinforcement Learning.** Behavior regularization can be achieved either implicitly or explicitly. Explicit methods usually necessitate the construction of a behavior model to regularize the learned policy. For example, TD3+BC (Fujimoto & Gu, 2021) implicitly views the behavior as Gaussians and introduces an auxiliary L2-loss term to realize the regularization. SBAC (Xu et al., 2021) and Fisher-BRC (Kostrikov et al., 2021) leverage explicit Gaussian (mixture) behavior models. BCQ (Fujimoto et al., 2019) and BEAR (Kumar et al., 2019) leverage VAE behavior models (Kingma & Welling, 2014). Diffusion-QL (Wang et al., 2023a) is similar to TD3+BC but swaps the auxiliary loss with a diffusion-centric objective. QGPO (Lu et al., 2023) views a pretrained diffusion behavior as the Bayesian prior in energy-guided sampling.

**Diffusion Models in Offline Reinforcement Learning.** Recent advancements in offline RL have identified diffusion models as an impactful tool. A primary strength of diffusion modeling lies in its robust generative capability, combined with a straightforward training pipeline (Dhariwal & Nichol, 2021; Xu et al., 2022; Poole et al., 2023). This renders it particularly suitable for modeling heterogeneous behavior datasets (Janner et al., 2022; Wang et al., 2023a; Ajay et al., 2023; Hansen-Estruch et al., 2023), generating in-support actions for Q-learning (Goo & Niekum, 2022; Lu et al., 2023), and representing multimodal policies (Chen et al., 2023; Pearce et al., 2022).

However, a significant concern for integrating diffusion models into RL is the considerable time taken for sampling. Various strategies have been proposed to address this challenge, including employing a parallel sampling scheme during both training and evaluation (Chen et al., 2023), utilizing a specialized diffusion ODE solver such as DPM-solver (Lu et al., 2022; 2023), adopting an approximate diffusion sampling scheme to minimize sampling steps required (Kang et al., 2023), and crafting high-throughput network architectures (Hansen-Estruch et al., 2023). While these techniques offer improvements, they don't entirely eliminate the need for iterative sampling.

**Score Distillation Methods.** Recent developments in text-to-3D generation have enabled the transformation of textual information into 3D content without requiring any 3D training data (Poole et al., 2023; Wang et al., 2023b). This is realized by distilling knowledge from text-to-image diffusion models. A representative method in this domain is DreamFusion (Poole et al., 2023). It optimizes a 3D NeRF model (Mildenhall et al., 2021) by ensuring its projected 2D gradient follows the score direction of a large-scale, pretrained 2D diffusion model (Saharia et al., 2022). Similar to DreamFusion, SRPO also employs a diffusion model to guide the training of a subsequent network. However, our method

| Dataset | Environment | BEAR | TD3+BC | IQL | SfBC | Diffuser | Diffusion-QL | QGPO | IDQL | SRPO (Ours) |
|---|---|---|---|---|---|---|---|---|---|---|
| Medium-Expert | HalfCheetah | 53.4 | 90.7 | 86.7 | **92.6** | 79.8 | **96.8** | 93.5 | 95.9 | **92.2 ± 3.0** |
| Medium-Expert | Hopper | 96.3 | 98.0 | 91.5 | **108.6** | 107.2 | **111.1** | 108.0 | 108.6 | 100.1 ± 13.9 |
| Medium-Expert | Walker2d | 40.1 | **110.1** | **109.6** | **109.8** | 108.4 | **110.1** | **110.7** | **112.7** | **114.0 ± 2.1** |
| Medium | HalfCheetah | 41.7 | 48.3 | 47.4 | 45.9 | 44.2 | 51.1 | 54.1 | 51.0 | **60.4 ± 0.8** |
| Medium | Hopper | 52.1 | 59.3 | 66.3 | 57.1 | 58.5 | 90.5 | **98.0** | 65.4 | **95.5 ± 2.0** |
| Medium | Walker2d | 59.1 | **83.7** | 78.3 | 77.9 | 79.7 | **87.0** | 86.0 | 82.5 | **84.4 ± 4.4** |
| Medium-Replay | HalfCheetah | 38.6 | 44.6 | 44.2 | 37.1 | 42.2 | 47.8 | 47.6 | 45.9 | **51.4 ± 3.4** |
| Medium-Replay | Hopper | 33.7 | 60.9 | 94.7 | 86.2 | **101.3** | **100.7** | 96.9 | 92.1 | **101.2 ± 1.0** |
| Medium-Replay | Walker2d | 19.2 | 81.8 | 73.9 | 65.1 | 61.2 | **95.5** | 84.4 | 85.1 | 84.6 ± 7.1 |
| **Average (Locomotion)** | | 51.9 | 75.3 | 76.9 | 75.6 | 75.3 | **88.0** | 86.6 | 82.1 | **87.1** |
| Default | AntMaze-umaze | 73.0 | 78.6 | 87.5 | 92.0 | - | 93.4 | **96.4** | 94.0 | **97.1 ± 2.7** |
| Diverse | AntMaze-umaze | 61.0 | 71.4 | 62.2 | **85.3** | - | 66.2 | 74.4 | 80.2 | **82.1 ± 10.8** |
| Play | AntMaze-medium | 0.0 | 10.6 | 71.2 | **81.3** | - | 76.6 | **83.6** | **84.5** | **80.7 ± 7.1** |
| Diverse | AntMaze-medium | 8.0 | 3.0 | 70.0 | **82.0** | - | 78.6 | **83.8** | **84.8** | 75.0 ± 12.3 |
| Play | AntMaze-large | 0.0 | 0.2 | 39.6 | 59.3 | - | 46.4 | **66.6** | 63.5 | 53.6 ± 12.5 |
| Diverse | AntMaze-large | 0.0 | 0.0 | 47.5 | 45.5 | - | 56.6 | **64.8** | **67.9** | 53.6 ± 6.3 |
| **Average (AntMaze)** | | 23.7 | 27.3 | 63.0 | 74.2 | - | 69.6 | 78.3 | **79.1** | 73.6 |

Table 1: Evaluation numbers of D4RL benchmarks (normalized as suggested by Fu et al. (2020)). We report mean ± standard deviation of algorithm performance across 6 random seeds at the end of training. Numbers within 5 % of the maximum in every individual task are highlighted.

emphasizes score regularization as opposed to score distillation. The behavior score is additionally incorporated to regularize the Q-gradient instead of being the only supervising signal.

## 6 EVALUATION

### 6.1 D4RL PERFORMANCE

In Table 1, we evaluate the D4RL performance (Fu et al., 2020) of SRPO against other offline RL algorithms. Our chosen benchmarks include conventional methods like BEAR (Kumar et al., 2019), TD3+BC (Fujimoto & Gu, 2021), and IQL (Kostrikov et al., 2022), which feature extracting a Gaussian/Dirac policy for evaluation. We also look at newer diffusion-based offline RL techniques, such as Diffuser (Janner et al., 2022), DIffusion-QL (Wang et al., 2023a), SfBC (Chen et al., 2023), QGPO (Lu et al., 2023), and IDQL (Hansen-Estruch et al., 2023). These methods tend to be more computationally intensive but generally offer better results.

Of all the baselines, our comparison with IDQL is particularly informative. This is because SRPO shares a virtually identical training pipeline and model architecture for critic and behavior models with IDQL, as is deliberately crafted. The most significant distinction lies in their approaches for extracting policy: IDQL skips the policy extraction step, choosing to evaluate directly with the behavior policy, using a selecting-from-behavior-candidates technique (Chen et al., 2023). In contrast, SRPO extracts a Dirac policy from behavior and critic models.

Overall, SRPO consistently surpasses referenced baselines that also utilize a Gaussian (or Dirac) inference policy, leading by large margins in the majority of tasks. It also comes close to matching the benchmarks set by other state-of-the-art diffusion-based methods, such as Diffusion-QL and IDQL, though it features a much simpler inference policy. Moreover, in the case of the convergence rate and training stability, we showcase training plots of SRPO alongside several baselines in Figure 6. Results also suggest that compared with both weighted regression methods like IQL and reverse-KL-based optimization strategies like Diffusion-QL, SRPO exhibits more favorable attributes. We attribute the performance gain of SRPO to the utilization of diffusion behavior modeling, which facilitates a more effective realization of the training formulation.

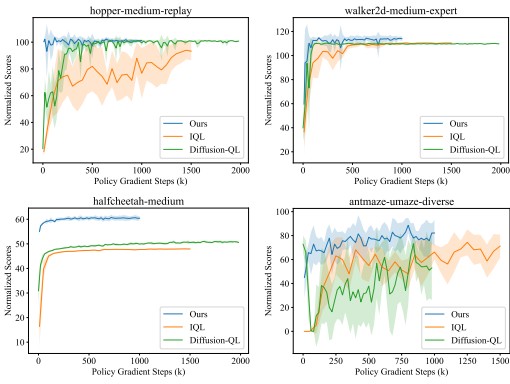

Figure 6: Training curves of SRPO (ours) and several baselines. See complete experimental results in Appendix D.

## 6.2 COMPUTATIONAL EFFICIENCY

As indicated by Figure 1 and 7, SRPO maintains high computational efficiency, especially fast inference speed, while enabling the use of a powerful diffusion model. Notably, the action sampling speed of SRPO is 25 to 1000 times faster than that of other diffusion-based methods. Additionally, its computational FLOPS amount to only 0.25% to 0.01% of other methods. This makes SRPO ideal for computation-sensitive contexts such as robotics. It also speeds up experimentation, since policy evaluation typically consumes over half of the experiment duration for previous diffusion-based approaches. This efficiency stems from SRPO's design, which completely avoids diffusion sampling throughout both training and evaluation procedures.

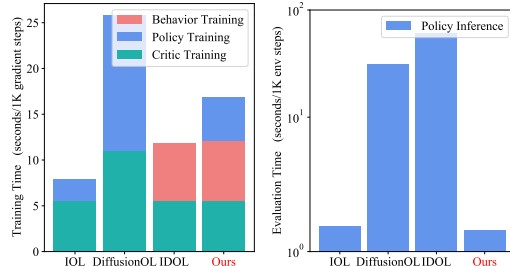

Figure 7: Training and inference (evaluation) time required for different algorithms in D4RL locomotion tasks. All experiments are conducted with the same PyTorch backend and the same computing hardware setup.

## 6.3 ABLATION STUDIES

**Weighting function** $\omega(t)$. As is explained in Section 4.2, if $\omega(t) \propto \delta(t - 0.02)$, the surrogate objective nearly recovers the original one, which could guarantee convergence to the wanted optimal policy $\pi^*$. Adjusting $\omega(t)$ to ensemble diffused behavior score at various times $t \in (0, 1)$ might yield a smoother and more robust gradient landscape. However, it biases the original training objective. In an extreme case where $\omega(t) \propto \delta(t - 0.98)$, the estimated behavior score almost becomes the score function of pure Gaussian distribution, making it entirely helpless. From Figure 8, we empirically observe that Antmaze tasks are sensitive to variation of $\omega(t)$, while Locomotion tasks are not. Overall, we find $\omega(t) = \sigma_t^2$ is a suitable choice for all tested tasks and choose it as the default hyperparameter throughout our experiments. This hyperparameter choice is consistent with previous literature (Poole et al., 2023).

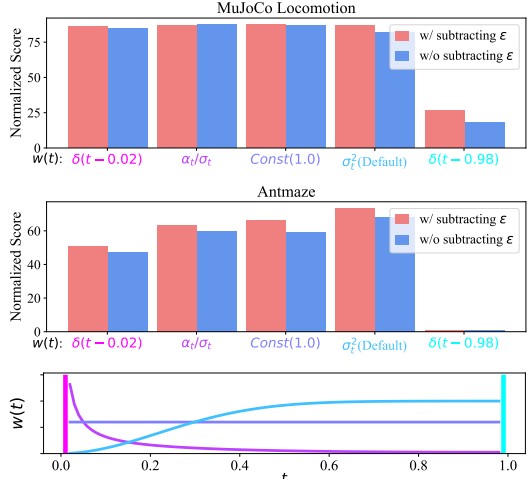

Figure 8: Ablation for implementation details of Eq. (15). All results are averaged under 6 random seeds.

**Subtracting baselines** $\epsilon$. In Figure 8, we show that subtracting baselines $\epsilon$ from the estimated behavior gradient, as is described in Eq. (15), consistently offers slightly better performance.

**Temperature coefficient** $\beta$ **and others.** Varying $\beta$ directly controls the conservativness of policy, and thus influences the performance. Experimental results with respect to $\beta$ and other implementation choices are presented in Appendix C.

## 7 CONCLUSION

In this paper, we introduce Score Regularized Policy Optimization (SRPO), an innovative offline RL algorithm harnessing the capabilities of diffusion models while circumventing the time-consuming diffusion sampling scheme. SRPO tackles the behavior-regularized policy optimization problem and provides a computationally efficient way to realize behavior regularization through diffusion behavior modeling. The fusion of SRPO with techniques like implicit Q-learning can further solidify its application in computationally sensitive domains, such as robotics.

## REPRODUCIBILITY

To ensure that our work is reproducible, we submit the source code as supplementary material. Reported experimental numbers are averaged under multiple random seeds. We provide implementation details of our work in Appendix C.

## ACKNOWLEDGEMENT

This work was supported by the National Key Research and Development Program of China (No. 2020AAA0106302), NSFC Projects (Nos. 62350080, 92370124, 92248303, 62106123, 61972224), BNRist (BNR2022RC01006), Tsinghua Institute for Guo Qiang, and the High Performance Computing Center, Tsinghua University. J.Z. was also supported by the New Cornerstone Science Foundation through the XPLORER PRIZE.

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

# A COMPLETE 2D EXPERIMENTS

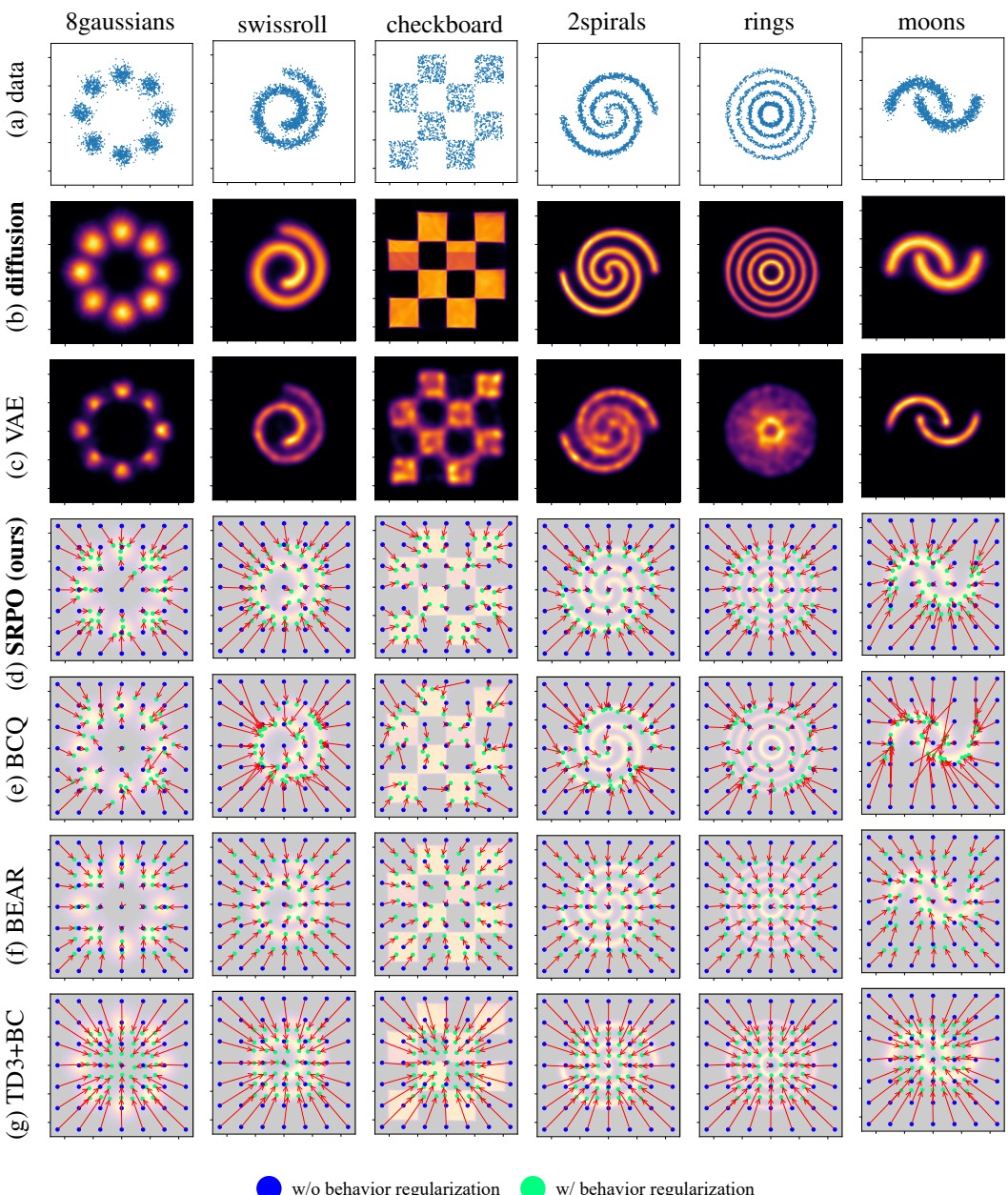

Figure 9: Experimental results of SRPO and other baselines in 2D bandit settings.

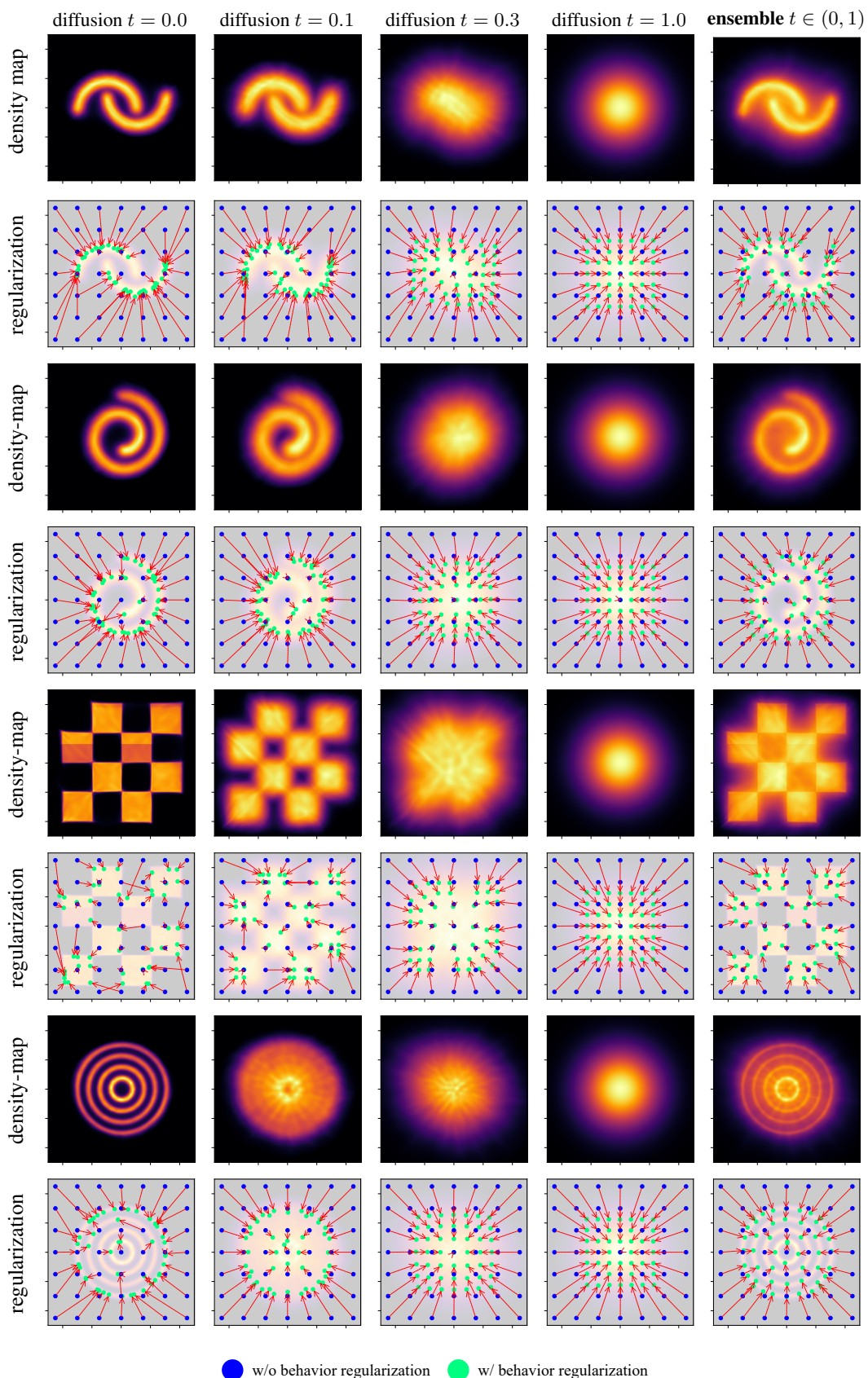

Figure 10: Empirical benefits of ensembling multiple diffusion times

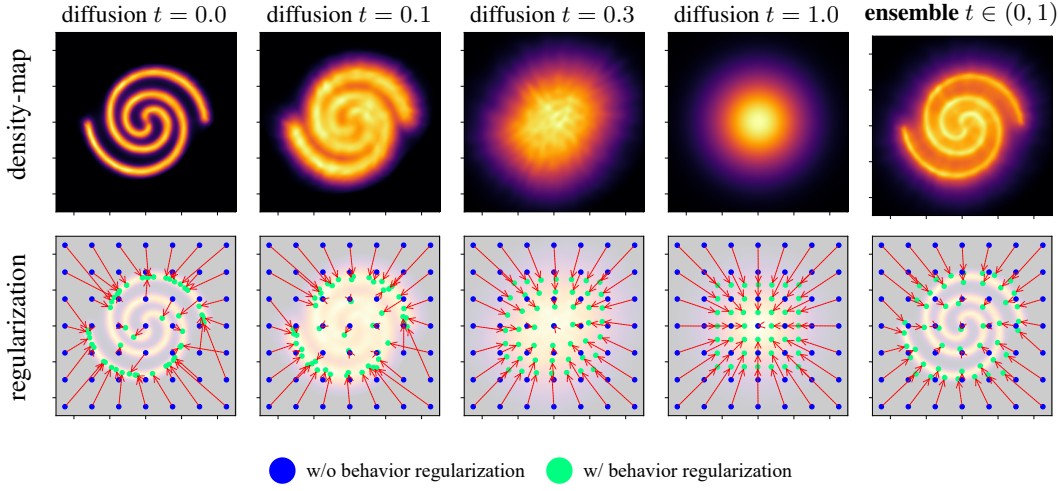

Figure 11: Empirical benefits of ensembling multiple diffusion times

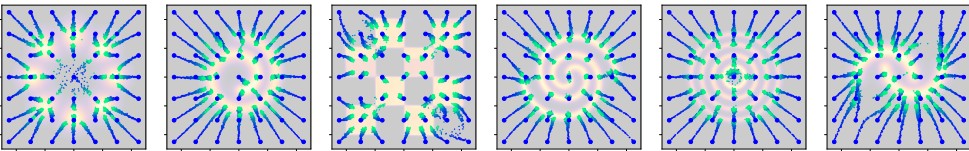

Figure 12: Effect of varying temperature $\beta$.

# B  THEORETICAL ANALYSIS

In this section, we provide some theoretical analysis related to the SPRO algorithm.

First, we aim to provide some insights into why it is reasonable to replace

$$\mathcal{L}_\pi(\theta) = \mathbb{E}_{s \sim \mathcal{D}^\mu, a \sim \pi_\theta} Q_\phi(s, a) - \frac{1}{\beta} D_{\mathrm{KL}} \left[ \pi_\theta(\cdot|s) || \mu(\cdot|s) \right]$$

with the surrogate objective

$$\mathcal{L}_\pi^{\mathrm{surr}}(\theta) = \mathbb{E}_{s, a \sim \pi_\theta} Q_\phi(s, a) - \frac{1}{\beta} \mathbb{E}_{t, s} \omega(t) \frac{\sigma_t}{\alpha_t} D_{\mathrm{KL}} \left[ \pi_{t, \theta}(\cdot|s) || \mu_t(\cdot|s) \right]. \tag{16}$$

**Proposition 1.** *Given that $\pi$ is sufficiently expressive, for any time $t$, any state $s$, we have*

$$\arg\min_\pi D_{\mathrm{KL}} \left[ \pi_t(\cdot|s) || \mu_t(\cdot|s) \right] = \arg\min_\pi D_{\mathrm{KL}} \left[ \pi(\cdot|s) || \mu(\cdot|s) \right],$$

*where both $\mu_t$ and $\pi_t$ follow the same predefined diffusion process in Eq. (4).*

*Proof.* Our proof is inspired by Wang et al. (2023b). Regarding the property of the KL divergence, we have $\arg\min_\pi D_{\mathrm{KL}} \left[ \pi(\cdot|s) || \mu(\cdot|s) \right] = \mu(\cdot|s)$ and $\pi_t^*(\cdot|s) := \arg\min_\pi D_{\mathrm{KL}} \left[ \pi_t(\cdot|s) || \mu_t(\cdot|s) \right] = \mu_t(\cdot|s)$.

We conclude that $\pi_t^*(\cdot|s) = \mu_t(\cdot|s)$ is equivalent to $\pi^*(\cdot|s) = \mu(\cdot|s)$ by transforming all distributions into their characteristic functions.

According to the forward diffusion process defined by Eq. (4), for any prespecified state $s$, we have

$$a_t^\pi = \alpha_t a_0^\pi + \sigma_t \epsilon = \alpha_t a^\pi + \sigma_t \epsilon,$$

such that

$$\pi_t(a_t|s) = \int \mathcal{N}(a_t | \alpha_t a, \sigma_t^2 I) \pi(a|s) \mathrm{d}a,$$

Therefore, the characteristic function of $\pi_t(a_t|s)$ satisfies

$$\phi_{\pi_t(\cdot|s)}(u) = \phi_{\pi(\cdot|s)}(\alpha_t u) \phi_{\mathcal{N}(\mathbf{0}, I)}(\sigma_t u) = \phi_{\pi(\cdot|s)}(\alpha_t s) e^{-\frac{\sigma_t^2 u^2}{2}}.$$

Similarly, we also have

$$\phi_{\mu_t(\cdot|s)}(u) = \phi_{\mu(\cdot|s)}(\alpha_t u) \phi_{\mathcal{N}(\mathbf{0}, I)}(\sigma_t u) = \phi_{\mu(\cdot|s)}(\alpha_t s) e^{-\frac{\sigma_t^2 u^2}{2}}.$$

Finally, we can see that

$$\pi_t^*(\cdot|s) = \mu_t(\cdot|s) \Leftrightarrow \phi_{\pi_t^*(\cdot|s)}(u) = \phi_{\mu_t(\cdot|s)}(u) \Leftrightarrow \phi_{\pi^*(\cdot|s)}(u) = \phi_{\mu(\cdot|s)}(u) \Leftrightarrow \pi^*(\cdot|s) = \mu(\cdot|s)$$

$\square$

**Remark 1.** *It is imperative to note that although $\arg\min_\pi D_{\mathrm{KL}} \left[ \pi_t(\cdot|s) || \mu_t(\cdot|s) \right]$ and $\arg\min_\pi D_{\mathrm{KL}} \left[ \pi(\cdot|s) || \mu(\cdot|s) \right]$ converge to the same global optimal solution. $\max \mathcal{L}_\pi^{surr}(\theta)$ does not converge to the wanted optimal policy $\pi^*(a|s) \propto \mu(a|s) \exp(\beta Q_\phi(s, a))$ while $\max \mathcal{L}_\pi(\theta)$ does. This discrepancy arises primarily from the inclusion of the additional Q-function term. Furthermore, in practical scenarios, the parameterized policy $\pi_\theta$ lacks expressivity, which hinders its ability to truly attain the global optimum. Nevertheless, ensembling a series of $t$ might be beneficial for the optimization to get to a better minimum in practice. Since the distribution $\mu(\cdot|s)$ is usually complex and high-dimensional, it is easy to become trapped in a local minimum while maximizing its likelihood. However, the distribution of $\mu_t(\cdot|s)$ gets smoother as $t$ gets larger.*

Next, we derive the gradient for $\mathcal{L}_\pi^{\mathrm{surr}}(\theta)$ under the condition that $\pi_\theta$ is a deterministic actor.

**Proposition 2.** *Given that $\pi_\theta$ is a deterministic actor ($a = \pi_\theta(s)$) and $\epsilon^*$ is an optimal diffusion behavior model ($\epsilon^*(a_t|s, t) = -\sigma_t \nabla_{a_t} \log \mu_t(a_t|s)$), the gradient for optimizing $\max_\theta \mathcal{L}_\pi^{surr}(\theta)$ in Eq. (16) satisfies*

$$\nabla_\theta \mathcal{L}_\pi^{surr}(\theta) = \left[ \mathbb{E}_s \nabla_a Q_\phi(s, a) |_{a = \pi_\theta(s)} - \frac{1}{\beta} \mathbb{E}_{t, s, \epsilon} \omega(t) (\epsilon^*(a_t|s, t) - \epsilon) |_{a_t = \alpha_t \pi_\theta(s) + \sigma_t \epsilon} \right] \nabla_\theta \pi_\theta(s)$$

*Proof.* According to the predefined forward diffusion process (Eq. (4)), for any state $s$, we have

$$\pi_{\theta,t}(\boldsymbol{a}_t|\boldsymbol{s}) = \int \mathcal{N}(\boldsymbol{a}_t|\alpha_t\boldsymbol{a}, \sigma_t^2\boldsymbol{I})\pi_\theta(\boldsymbol{a}|\boldsymbol{s})\mathrm{d}\boldsymbol{a} = \int \mathcal{N}(\boldsymbol{a}_t|\alpha_t\boldsymbol{a}, \sigma_t^2\boldsymbol{I})\delta(\boldsymbol{a}-\pi_\theta(\boldsymbol{s}))\mathrm{d}\boldsymbol{a} = \mathcal{N}(\boldsymbol{a}_t|\alpha_t\pi_\theta(\boldsymbol{s}), \sigma_t^2\boldsymbol{I}).$$

Therefore $\pi_{\theta,t}(\cdot|\boldsymbol{s})$ is a Gaussian with expected value $\alpha_t\pi_\theta(\boldsymbol{s})$ and variance $\sigma_t^2\boldsymbol{I}$. We rewrite the training objective below:

$$\mathcal{L}_\pi^{\text{surr}}(\theta) = \mathbb{E}_{\boldsymbol{s},\boldsymbol{a}\sim\pi(\cdot|\boldsymbol{s})}Q_\phi(\boldsymbol{s},\boldsymbol{a}) - \frac{1}{\beta}\mathbb{E}_{t,\boldsymbol{s}}\omega(t)\frac{\sigma_t}{\alpha_t}D_{\text{KL}}\left[\pi_{t,\theta}(\cdot|\boldsymbol{s})||\mu_t(\cdot|\boldsymbol{s})\right]$$

$$= \mathbb{E}_{\boldsymbol{s},\boldsymbol{a}\sim\pi(\cdot|\boldsymbol{s})}Q_\phi(\boldsymbol{s},\boldsymbol{a}) + \frac{1}{\beta}\mathbb{E}_{t,\boldsymbol{s},\boldsymbol{a}_t\sim\pi_{t,\theta}(\cdot|\boldsymbol{s})}\omega(t)\frac{\sigma_t}{\alpha_t}\left[\log\mu_t(\boldsymbol{a}_t|\boldsymbol{s}) - \log\pi_{t,\theta}(\boldsymbol{a}_t|\boldsymbol{s})\right]$$

$$= \mathbb{E}_{\boldsymbol{s}}Q_\phi(\boldsymbol{s},\boldsymbol{a})|_{\boldsymbol{a}=\pi_\theta(\boldsymbol{s})} + \frac{1}{\beta}\mathbb{E}_{t,\boldsymbol{s}}\omega(t)\frac{\sigma_t}{\alpha_t}\mathbb{E}_{\boldsymbol{a}_t\sim\mathcal{N}(\cdot|\alpha_t\pi_\theta(\boldsymbol{s}),\sigma_t^2\boldsymbol{I})}\left[\log\mu_t(\boldsymbol{a}_t|\boldsymbol{s}) - \log\pi_{t,\theta}(\boldsymbol{a}_t|\boldsymbol{s})\right]$$

Then we derive the gradient of $\mathcal{L}_\pi^{\text{surr}}(\theta)$ by applying the chain rule and the parameterization trick:

$$\nabla_\theta\mathcal{L}_\pi^{\text{surr}}(\theta) = \frac{\partial\mathbb{E}_{\boldsymbol{s}}Q_\phi(\boldsymbol{s},\boldsymbol{a})}{\partial\boldsymbol{a}}\Big|_{\boldsymbol{a}=\pi_\theta(\boldsymbol{s})}\frac{\partial\pi_\theta(\boldsymbol{s})}{\partial\theta} + \frac{1}{\beta}\mathbb{E}_{t,\boldsymbol{s}}\omega(t)\frac{\sigma_t}{\alpha_t}\mathbb{E}_\epsilon\underbrace{\frac{\partial[\log\mu_t(\boldsymbol{a}_t|\boldsymbol{s})]}{\partial\boldsymbol{a}_t}}_{\text{behavior score}}\frac{\partial\boldsymbol{a}_t}{\partial\theta}\Big|_{\boldsymbol{a}_t=\alpha_t\pi_\theta(\boldsymbol{s})+\sigma_t\epsilon}$$

$$- \frac{1}{\beta}\mathbb{E}_{t,\boldsymbol{s}}\omega(t)\frac{\sigma_t}{\alpha_t}\left[\mathbb{E}_\epsilon\underbrace{\frac{\partial[\log\pi_{t,\theta}(\boldsymbol{a}_t|\boldsymbol{s})]}{\partial\boldsymbol{a}_t}}_{\text{policy score}}\frac{\partial\boldsymbol{a}_t}{\partial\theta}\Big|_{\boldsymbol{a}_t=\alpha_t\pi_\theta(\boldsymbol{s})+\sigma_t\epsilon} + \underbrace{\mathbb{E}_{\boldsymbol{a}_t\sim\pi_{t,\theta}(\cdot|\boldsymbol{s})}\frac{\partial\log\pi_{t,\theta}(\boldsymbol{a}_t|\boldsymbol{s})}{\partial\theta}}_{\text{parameter score}}\right] \quad (17)$$

The behavior score term in the above equation can be represented by the optimal diffusion behavior model:

$$\frac{\partial[\log\mu_t(\boldsymbol{a}_t|\boldsymbol{s})]}{\partial\boldsymbol{a}_t} = -\frac{\boldsymbol{\epsilon}^*(\boldsymbol{a}_t|\boldsymbol{s},t)}{\sigma_t}.$$

The policy score term is the score function of $\pi_{t,\theta}(\cdot|\boldsymbol{s}) = \mathcal{N}(\alpha_t\pi_\theta(\boldsymbol{s}), \sigma_t^2\boldsymbol{I})$, we have

$$\frac{\partial[\log\pi_{t,\theta}(\boldsymbol{a}_t|\boldsymbol{s})]}{\partial\boldsymbol{a}_t} = -\frac{\partial}{\partial\boldsymbol{a}_t}\frac{\|\boldsymbol{a}_t - \alpha_t\pi_\theta(\boldsymbol{s})\|_2^2}{2\sigma_t^2}\Big|_{\boldsymbol{a}_t=\alpha_t\pi_\theta(\boldsymbol{s})} = \frac{\epsilon}{\sigma_t}.$$

The parameter score term equals 0 for any distribution $\pi_{t,\theta}$, regardless of whether it is Gaussian:

$$\mathbb{E}_{\boldsymbol{a}_t\sim\pi_{t,\theta}(\cdot|\boldsymbol{s})}\frac{\partial\log\pi_{t,\theta}(\boldsymbol{a}_t|\boldsymbol{s})}{\partial\theta} = \int\pi_{t,\theta}(\boldsymbol{a}_t|\boldsymbol{s})\frac{\partial\log\pi_{t,\theta}(\boldsymbol{a}_t|\boldsymbol{s})}{\partial\theta}\mathrm{d}\boldsymbol{a}_t$$

$$= \int\frac{\partial\pi_{t,\theta}(\boldsymbol{a}_t|\boldsymbol{s})}{\partial\theta}\mathrm{d}\boldsymbol{a}_t$$

$$= \frac{\partial}{\partial\theta}\int\pi_{t,\theta}(\boldsymbol{a}_t|\boldsymbol{s})\mathrm{d}\boldsymbol{a}_t$$

$$= 0$$

Continue on $\nabla_\theta\mathcal{L}_\pi^{\text{surr}}(\theta)$ and substitute the conclusions from above:

$$\nabla_\theta\mathcal{L}_\pi^{\text{surr}}(\theta) = \frac{\partial\mathbb{E}_{\boldsymbol{s}}Q_\phi(\boldsymbol{s},\boldsymbol{a})}{\partial\boldsymbol{a}}\Big|_{\boldsymbol{a}=\pi_\theta(\boldsymbol{s})}\frac{\partial\pi_\theta(\boldsymbol{s})}{\partial\theta} + \frac{1}{\beta}\mathbb{E}_{t,\boldsymbol{s}}\omega(t)\frac{\sigma_t}{\alpha_t}\mathbb{E}_\epsilon - \frac{\boldsymbol{\epsilon}^*(\boldsymbol{a}_t|\boldsymbol{s},t)}{\sigma_t}\frac{\partial\boldsymbol{a}_t}{\partial\theta}\Big|_{\boldsymbol{a}_t=\alpha_t\pi_\theta(\boldsymbol{s})+\sigma_t\epsilon}$$

$$- \frac{1}{\beta}\mathbb{E}_{t,\boldsymbol{s}}\omega(t)\frac{\sigma_t}{\alpha_t}\left[\mathbb{E}_\epsilon\frac{\epsilon}{\sigma_t}\frac{\partial\boldsymbol{a}_t}{\partial\theta}\Big|_{\boldsymbol{a}_t=\alpha_t\pi_\theta(\boldsymbol{s})+\sigma_t\epsilon} + 0\right]$$

$$= \frac{\partial\mathbb{E}_{\boldsymbol{s}}Q_\phi(\boldsymbol{s},\boldsymbol{a})}{\partial\boldsymbol{a}}\Big|_{\boldsymbol{a}=\pi_\theta(\boldsymbol{s})}\frac{\partial\pi_\theta(\boldsymbol{s})}{\partial\theta} - \frac{1}{\beta}\mathbb{E}_{t,\boldsymbol{s}}\omega(t)\frac{1}{\alpha_t}\mathbb{E}_\epsilon\left[\boldsymbol{\epsilon}^*(\boldsymbol{a}_t|\boldsymbol{s},t) - \epsilon\right]\frac{\partial\boldsymbol{a}_t}{\partial\theta}\Big|_{\boldsymbol{a}_t=\alpha_t\pi_\theta(\boldsymbol{s})+\sigma_t\epsilon}$$

$$= \frac{\partial\mathbb{E}_{\boldsymbol{s}}Q_\phi(\boldsymbol{s},\boldsymbol{a})}{\partial\boldsymbol{a}}\Big|_{\boldsymbol{a}=\pi_\theta(\boldsymbol{s})}\frac{\partial\pi_\theta(\boldsymbol{s})}{\partial\theta} - \frac{1}{\beta}\mathbb{E}_{t,\boldsymbol{s}}\omega(t)\frac{1}{\alpha_t}\mathbb{E}_\epsilon\left[\boldsymbol{\epsilon}^*(\boldsymbol{a}_t|\boldsymbol{s},t) - \epsilon\right]\alpha_t\frac{\partial\pi_\theta(\boldsymbol{s})}{\partial\theta}\Big|_{\boldsymbol{a}_t=\alpha_t\pi_\theta(\boldsymbol{s})+\sigma_t\epsilon}$$

$$= \left[\mathbb{E}_{\boldsymbol{s}}\nabla_{\boldsymbol{a}}Q_\phi(\boldsymbol{s},\boldsymbol{a})|_{\boldsymbol{a}=\pi_\theta(\boldsymbol{s})} - \frac{1}{\beta}\mathbb{E}_{t,\boldsymbol{s},\epsilon}\omega(t)(\boldsymbol{\epsilon}^*(\boldsymbol{a}_t|\boldsymbol{s},t)\underbrace{-\epsilon}_{\text{subtracted baseline}})|_{\boldsymbol{a}_t=\alpha_t\pi_\theta(\boldsymbol{s})+\sigma_t\epsilon}\right]\nabla_\theta\pi_\theta(\boldsymbol{s})$$

$\square$

**Remark 2.** *The subtracted baseline $\epsilon$ above corresponds to the policy score term in Eq. (17). It does not influence the expected value of the empirical surrogate gradient $\nabla_\theta \mathcal{L}_\pi^{surr}(\theta)$. To see this, consider isolating the baseline term $\frac{1}{\beta}\mathbb{E}_{t,s,\epsilon}\epsilon$. Given that $\epsilon$ is a random Gaussian noise independent of both state $s$ and time $t$, we can prove $\mathbb{E}_{t,s,\epsilon}\epsilon = 0$. Still, previous work (Roeder et al., 2017; Poole et al., 2023) shows that keeping the policy score term can reduce the variance of the gradient estimate and thus speed up training.*

## C    Experimental Details for D4RL Benchmarks

**Critic training.**    We train our critic models following Kostrikov et al. (2022). For the convenience of readers, we recap some key hyperparameters: All networks are 2-layer MLPs with 256 hidden units and ReLU activations. We train them for 1.5M gradient steps using Adam optimizer with a learning rate of 3e-4. Batchsize is 256. Temperature: $\tau = 0.7$ (MuJoCo locomotion) and $\tau = 0.9$ (Antmaze).

**Behavior training.**    We adopt the model architecture proposed by Hansen-Estruch et al. (2023), with a modification to accommodate continuous-time input. A single scalar time input is mapped to a high-dimensional feature using Gaussian Fourier Projection before concatenated with other inputs. The network is basically a 6-layer MLP with residual connections, layer normalizations, and dropout regularizations. We train the behavior model for 2.0M gradient steps using AdamW optimizer with a learning rate of 3e-4 and a batchsize of 2048. Empirical observations suggest that much fewer pretraining iterations (e.g., 0.5M steps) do not cause a drastic performance drop, but we want to ensure training convergence in this work. The diffusion data perturbation method follows the default VPSDE setting in Song et al. (2021) and is consistent with prior work (Lu et al., 2023).

**Policy extraction (Locomotion).**    The policy model is a 2-layer MLP with 256 hidden units and ReLU activations. It is trained for 1.0M gradient steps using Adam optimizer with a learning rate of 3e-4 and a batchsize of 256. For all tasks $\omega(t) = \sigma_t^2$. For the temperature coefficient, we sweep over $\beta \in \{0.01, 0.02, 0.05, 0.1, 0.2, 0.5\}$ and observe large variances in appropriate values across different tasks (Figure 15). We speculate this might be due to $\beta$ being closely intertwined with the behavior distribution and the variance of the Q-value. These factors might exhibit entirely different characteristics across diverse tasks. Our choices for $\beta$ are detailed in Table 2.

**Policy extraction (Antmaze).**    We empirically find that a deeper policy network improves overall performance in Antmaze tasks (Figure 14). As a result, we employ a 4-layer MLP as the policy model. Additionally, we observe that the adopted implicit Q-learning method sometimes has a training instability issue in Antmaze tasks, resulting in highly divergent estimated Q-values (Figure 13). To stabilize training for policy extraction, we replace the temperature coefficient $\beta$ with $\beta_{\text{norm}}(s, a) := \frac{\beta}{\|\nabla_a Q(s,a)\|_2}$. We sweep over $\beta \in \{0.01, 0.02, ..., 0.05\}$ for umaze environments, and $\beta \in \{0.03, 0.04, ..., 0.08\}$ for other environments. Our choices for $\beta$ are detailed in Table 2. Other hyperparameters remain consistent with those in the locomotion tasks.

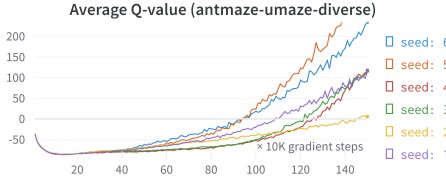

Figure 13: The training instability issue.

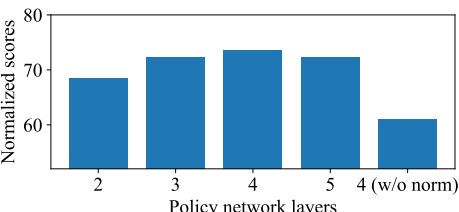

Figure 14: Ablation for Antmaze tasks.

**Evaluation.**    We run all experiments over 6 independent trials. For each trial, we additionally collect the evaluation score averaged across 20 test episodes at regular intervals for plots in Figure 16. The average performance at the end of training is reported in Table 1. We use NVIDIA A40 GPUs for reporting computing results in Figure 1.

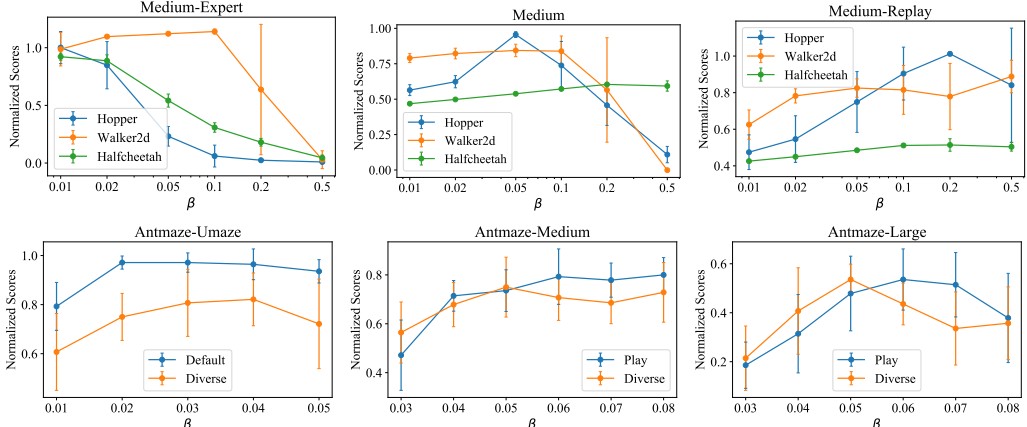

Figure 15: Ablation of the temperature coefficient $\beta$ in D4RL benchmarks.

| | Walker2d | Halfcheetah | Hopper |
|---|---|---|---|
| Locomotion-Medium-Expert | 0.1 | 0.01 | 0.01 |
| Locomotion-Medium | 0.05 | 0.2 | 0.05 |
| Locomotion-Medium-Replay | 0.5 | 0.2 | 0.2 |
| | Umaze | Medium | Large |
| AntMaze-Fixed | 0.02 | 0.08 | 0.06 |
| AntMaze-Diverse | 0.04 | 0.05 | 0.05 |

Table 2: Temperature coefficient $\beta$ for every individual task.

# D TRAINING CURVES FOR OFFLINE REINFORCEMENT LEARNING

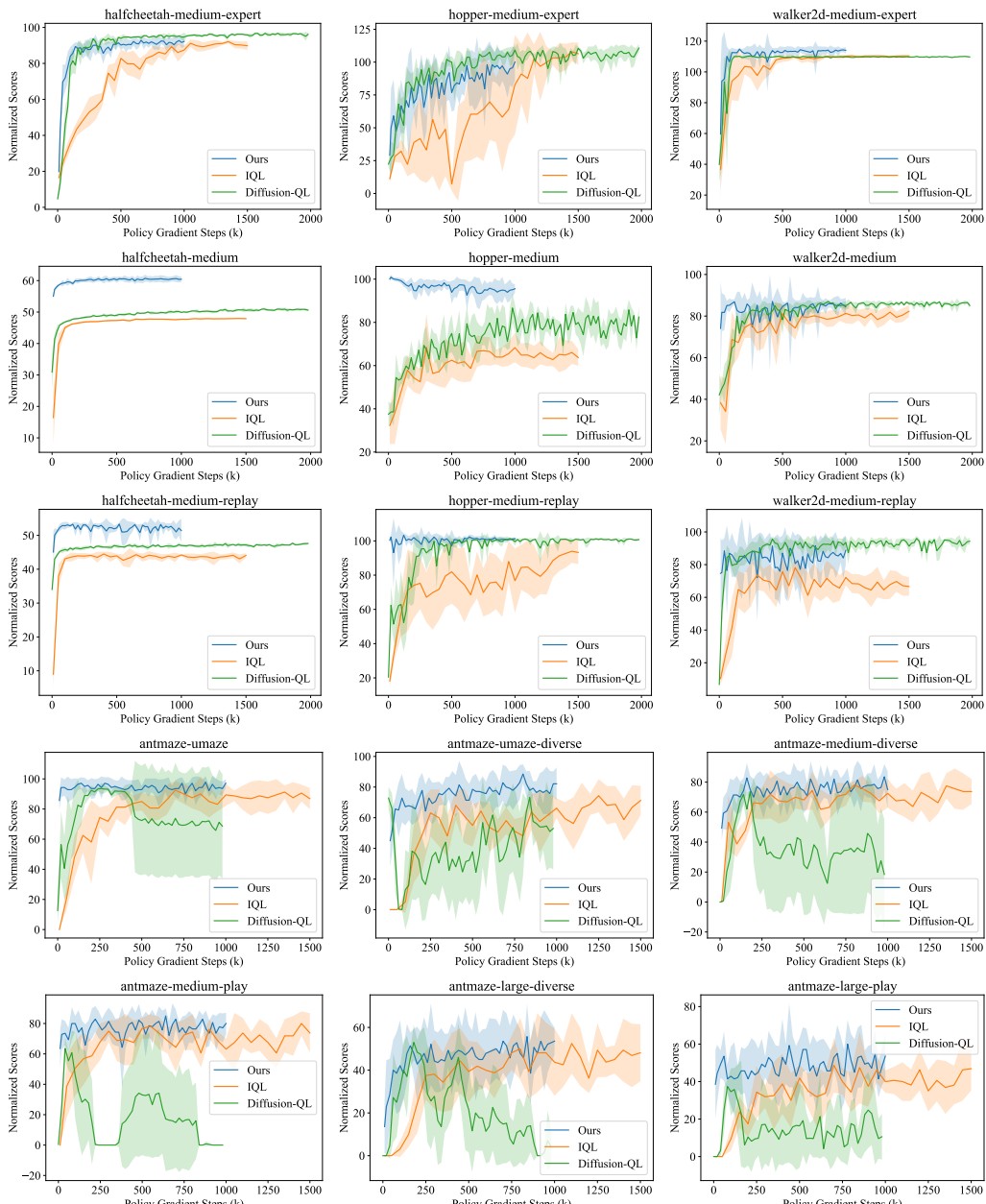

Figure 16: Training curves of SRPO (ours) and several baselines. Scores are normalized according to Fu et al. (2020).

