# OpenReview forum: "Score Regularized Policy Optimization through Diffusion Behavior"
_ICLR.cc/2024/Conference — ICLR 2024 poster_

### Official Review · Reviewer_f3qQ · 2023-10-19

**Soundness:** 3 good
**Presentation:** 2 fair
**Contribution:** 3 good
**Rating:** 6
**Confidence:** 3

**Summary:**

This paper aims to improve the inference efficiency of diffusion models in the context of offline reinforcement learning, by proposing a deterministic inference policy from critic models and pretrained diffusion behavior models. The pretrained diffusion behavior models are then utilized to regularize the policy gradient. Experiments are conducted on D4RL in locomotion tasks, the speed is improved without an evident performance drop.

**Strengths:**

- The research problem of improving inference efficiency for diffusion models in offline reinforcement learning is interesting and worth investigating.

- The paper is well organized and written in general, with motivations and methods well explained (but somewhat unclear to me).

- The experiments show evident improvement in time cost and demonstrate the effectiveness.

**Weaknesses:**

- Some implementation details are unclear, while the Fig.1 shows the comparison of the inference efficiency, the actual time cost for training is not mentioned. (In Appendix C, it seems the actual time used for training is not specified?)

- I get a bit confused by some high-level assumptions in this work. Essentially, I believe there are two types of distributions that matter in the context of Diffusion models for offline RL, namely the actual data distribution and the learnable Gaussian kernel (noise) distribution (which is pre-scheduled on the mean values with fixed variance). If the actual learnable $\pi_\theta$ is used as the appropriation/predictor of the policy distribution (which I believe is the case according to Eq. 9?), then what is the difference between this proposed paradigm and the Gaussian case, while the latter is believed to be lacking in expressivity, as mentioned by the authors in the intro and Fig.2, however the $\pi_\theta$ that you are sampling from is also a Gaussian with known variance and learnable mean value.

- Following my previous point, which I think is also related to the choice of $t$ in Sec. 4.2 and also in Eq. 9. If I misunderstood the actual usage of distribution here, and the proposed SRPO method actually uses the actual distribution, which corresponds to $x_0$ (or somewhere close to $x_0$), as the optimal policy that you are trying to sample from, then the expressivity concern does not exist, but then in this case, the entropy term in Eq. 9 does not hold, because we don’t have any prior on the actual distribution of $x_0$ also on its variance?

**Questions:**

Please see the weaknesses for details. Overall, I think I get confused by the distributions from the DMs and the actual deployment in the context of RL in this work, as well as the rationale/motivation behind it.

---

> ### Author Response · Authors · 2023-11-16
> **Official Response to Reviewer f3qQ (1/2)**
>
> We sincerely thank reviewer f3qQ for his interest and detailed feedback on our work. His valuable feedback has significantly contributed to enhancing the clarity and overall presentation of our paper. **We have updated the manuscript with more experimental illustrations and improved presentations** (particularly in section 3).
>
> **Q1: The actual time cost for training is not mentioned (in Appendix C).**
>
> **A1:** The actual time cost for training was actually provided in Section 6.2 of the paper (Figure 7), where we compared training and evaluation time between our work and several baselines in detail. We present Figure 7 in tabular form below:
>
> | Algorithm    | Training Time (h)  | Evaluation Time (h) | Total Time (h)  | Evaluation/Total (\%)  |
> |--------------|:-----------------:|:-------------------:|:-------------------:|:-----------------:|
> | IQL          | 4.36            | 0.40               | 4.76                 | 8.4            |
> | DiffusionQL  | 14.3            | 8.66               | 22.96                 | 37.7            |
> | IDQL         | 6.57            | 16.9               | 23.47                |  72.0            |
> | SRPO(Ours)   | 9.32            | 0.38               | 9.7                |  3.9            |
>
> (Suppose each experiment requires 2,000K gradient steps for training, 50(times)*20(seeds) episodes of evaluation, test on Nvidia A40 GPUs)
>
>
> **Q2: Overall, I think I get confused by the distributions from the DMs and the actual deployment in the context of RL in this work**
>
> **A2:** We appreciate the opportunity to clarify the confusion, primarily regarding the definitions of $\pi_\theta(s)$ and $\pi_{\theta, t}(\cdot|s)$. **We have updated the manuscript, adding Figure 5 and Figure 10 to help better illustrate our idea.** Below is a summarization of the main distributions from the Diffusion Models:
>
>
> |   Diffusion time  |  &emsp;     $t=0$                       |       | &emsp;&emsp;&emsp;&emsp; $t \in (0,1)$  |
> |:--------:|:---------:|:-----:|:-----:|
> |                       |           |         diffusion (add noise)       |               |
> | behavior data:  |  $ \mu(a\|s)   $ | ----------------------------------> | $\mu_t(a_t\|s)$ |
> |                       |           |                |          ↓     |               |
> |  *diffusion''modeling*    |           |               |         supervise(Eq.13)      |
> |                       |           |                |         ↓      |               |
> |behavior model:  |  $ \epsilon_\psi(a\|s)   $ |  | $\epsilon_\psi(a_t\|s,t)$ |
> |                       |↓  |         <--------------  +$Q_\phi(s,a)$  **-------------->**   |        **↓**      |               |
> |  *score''regularization*    |        supervise(Eq.10)   |               |         **supervise(Eq.15)**      |
> |                       |         ↓  |                |         ↓      |               |
> | policy model:  |  $ \pi_\theta(s)   $ | **------------(define by diffusion)---------->** | **$\pi_{\theta, t}(\cdot\|s) $:=$\mathcal{N}(\cdot\|\alpha\_t \pi_\theta(s),\sigma_t^2 I ) $** |
> |                       |           |      **<---------------(supervise)---------------**           |               |               |
>
>
> (The method proposed in Section 4.2 of the paper is highlighted in **boldface**.)
>
> - $\mu(a|s)$: The underlying behavior policy for behavior dataset $D^{\mu}$.
> - $\mu_t(a|s)$: The noisy version of $\mu(a|s)$, defined by the forward diffusion process (Eq.4 in paper).
> - $\epsilon\_\psi(a\_t|s,t)$: The diffusion behavior model learned to represent $\nabla_a \log \mu\_t(a|s)$ for a series of $t\in(0,1)$. We need $\epsilon\_\psi$ to regularize gradient when optimizing policy $\pi\_\theta$.
> - $\pi_\theta(s)$: A deterministic policy which is actually **used for evaluation**.
> - $\pi\_{\theta, t}(\cdot|s)$: The noisy (diffused) version of $\pi\_\theta(s)$. **Note that** $\pi\_{\theta, t}(\cdot|s)$ **is defined by** $\pi\_\theta(s)$. Its mean is $\alpha\_t \pi\_\theta(s)$. It has fixed variance. As a result, **when we optimize $\pi\_{\theta, t}(\cdot|s)$, we are essentially optimizing the underlying evaluation policy $\pi\_\theta(s)$.** $\pi\_{\theta, t}(\cdot|s)$ is **not** used for evaluation.
>
>
> **Why optimize $\pi\_{\theta, t}(\cdot|s)$ as in Eq.15? Why not just directly optimize $\pi\_\theta(s)$ as in Eq.10?** (since we do not actually use $\pi\_{\theta, t}(\cdot|s)$)
>
> 1. Our goal is to leverage the pretrained diffusion model $\epsilon_\psi(a_t|s,t)$ at every $t \in (0,1)$, not just at $t=0$ (Eq. 15). Proposition 1 in our paper (proof in Appendix B) demonstrates the equivalence between minimizing $KL(\pi_\theta(s) || \mu(a|s))$ and minimizing $KL(\pi_{\theta, t}(\cdot|s) || \mu_t(a|s))$.
> 2. When $t$ is always 0, Eq.15 and Eq.10 become identical, making Eq.10 a special case of Eq.15.
> 3. Utilizing pretrained diffusion models across all $t \in (0,1)$ empirically improves performance. See Section 6.3 for ablations, and see Figure 5,10 for illustrations.

---

> ### Author Response · Authors · 2023-11-16
> **Official Response to Reviewer f3qQ (2/2)**
>
> **Q3: The final policy that you are sampling from is also a Gaussian, which is believed to be lacking in expressivity.  So what is the point of introducing diffusion models?**
>
> **A3:**
> While the diffusion **behavior** model $\mu_\psi(a|s)$, does **not** enhance the expressivity of the final evaluation policy $\pi\_\theta$, they better represent the behavior distribution $\mu(a|s)$. This **leads to more accurate behavior regularization and, consequently, better learning of $$\pi_\theta$**.**
>
>
> Empirically, our method significantly outperforms traditional behavior regularization methods like BCQ, BEAR, and BRAC in the D4RL benchmarks, as shown in Table 1 of the paper. These methods use VAE models for behavior modeling, which we believe cannot fully capture complex behavior policies.
>
> **We've also conducted additional qualitative experiments in 2D settings to support our claims**, comparing our method with previous ones and assessing the generative modeling capabilities of VAEs versus Diffusion. The results are included in the newly added Figures 4 and 9 of the paper.
>
>
> **Q4:The rationale/motivation behind our work. (high-level assumptions?)**
>
> **A4:** Our method aims to effectively and efficiently capture the "mode" of $\pi^*$, focusing on training a deterministic or Gaussian policy $\pi^{\text{gauss}}\_{\theta} (s) = \arg \max\_a \mu(a|s) \ e^{\beta Q(s, a)} $ **instead of a diverse one** $\pi_\theta^{\text{diff}}(a|s) \propto \mu(a|s) \ e^{\beta Q(s, a)}$.
>
>
> If there are two (multiple) optimal policies for one single task, our method will simply choose either one of them deterministically. Our assumption is that in many scenarios, inference speed is of much higher priority than policy diversity. For instance, in robotics, we often do not care about how many ways our robot can solve a task. Contrastively, we do care about whether our policy is optimal and efficient (we need high control frequency!). Also, current evaluation metrics of the D4RL benchmark only emphasize policy optimality and ignore policy diversity.
>
> **Q5: If I misunderstood the actual usage of distribution here, ... in this case, the entropy term in Eq. 9 does not hold.**
>
> **A5:** As clarified in **A4**, our evaluation policy is either deterministic or Gaussian, so the reviewer's understanding of its usage is correct. Therefore, the entropy term in Eq. 9 remains valid.

---

> > ### Comment · Reviewer_f3qQ · 2023-11-18
> > **Thanks for the rebuttal**
> >
> > I appreciate the authors' efforts in rebuttal. I think the diffusion modeling part seems clearer to me after reading the responses, and I am open to leaving the other reviewers to assess the RL part.

---

> > > ### Author Response · Authors · 2023-11-18
> > > **We are glad that our response helps**
> > >
> > > We are glad to see that we can clear the reviewer's confusion and clarify our motivation! If the reviewer has further questions or suggestions, we would be very happy to address them.

---

### Official Review · Reviewer_cVKT · 2023-10-24

**Soundness:** 2 fair
**Presentation:** 4 excellent
**Contribution:** 3 good
**Rating:** 8
**Confidence:** 3

**Summary:**

The paper addresses the slow inference issue of diffusion policies and proposes an offline RL algorithm to avoid iterative diffusion sampling process during policy evaluation by leveraging a critic model and a pretrained diffusion model.
The proposed method, SRPO, consists of three components: a critic model realized by the IQL, a pretrained diffusion model to explicitly model the behavior policy, and a policy extraction module utilizing the pretrained behavior model to regularize the policy gradient.
Experimental results on D4RL benchmark suggest the proposed method can achieve higher computational efficiency especially during inference compared to other diffusion-based policies, while maintaining comparable performance by exploiting the modeling expressiveness of diffusion models.
The main contribution of this paper is to employ a pretrained diffusion behavior model, which can approximate the score function of the behavior distribution in offline datasets, to regularize the policy gradient during the optimization of the actor.

**Strengths:**

1. The paper clearly states its motivation and presents a clear illustration to demonstrate the derivation of the proposed method, SRPO.
2. The paper provides reproducible details for its experiments, and make a relatively comprehensive comparison with both conventional behavior regularization methods and recent diffusion-based policies in offline RL, in terms of task performance and computational efficiency.

**Weaknesses:**

1. The paper mainly aims to improve the computational efficiency of diffusion-based polices, which is highlighted in computation-sensitive contexts such as robotics as stated in the paper, yet there is no experiment concerning the robot scenarios especially with real data. If such experimental results are provided, the central claim made by the paper can be more convincing.
2. The novelty of the proposed method is limited as it seems to be a combination of previous work, and especially an incremental work based on IDQL (Hansen-Estruch et al., 2023), though it has been compared in the experiments. The main difference between the two work is the policy extraction process, which is the main contribution of this paper but does not provide adequate contribution to the community.

**Questions:**

1. In Table 1, where do the results of baselines come from? Are they reported from original papers or all have been re-implemented for comparison？
2. In section 6.2, the paper claims that SRPO "completely avoids diffusion sampling throughout both training and evaluation procedures". But it is considered that the pretrained diffusion model for behavior policy still requires iterative sampling during the pretraining phase, which is also visualized in Fig 5 when compared to IDQL. This claim needs further explanation.
3. In Fig 5, are the results averaged across all locomotion tasks?

---

> ### Author Response · Authors · 2023-11-16
> **Official Response to Reviewer cVKT (1/2)**
>
> We thank the reviewer for his valuable feedback. We seek to address the reviewer's concern by expanding experimental evaluations and clarifying the contributions of our paper.
>
>
>
>
> **Q1: The novelty of the proposed method is limited as it seems to be a combination of previous work, and especially an incremental work based on IDQL.**
>
> **A1** We respectfully disagree with this assessment for several reasons:
>
> 1. **Our method is not a combination of previous works.** We propose score regularization, which leverages a pretrained diffusion model to **regularize optimization gradients instead of loss** during policy training. To the best of our knowledge, this is a novel and unique optimization strategy. Prior to our work, diffusion models in RL were limited to direct action sampling in an iterative and time-intensive manner. Our approach marks a significant departure from this.
> 2. **Our proposed score-regularization method is a general policy-optimization method instead of being incremental to IDQL.** While we have followed the framework of IDQL and SfBC for behavior modeling and IQL (or IDQL) for critic training to aid comparison and understanding, our score-regularization method is a broadly applicable policy optimization technique, not merely an extension of IDQL. Plus, the training of critics and behavior models is not claimed as a contribution in our paper.
> 3. **We have conducted additional experiments combining our method with other diffusion-RL methods (diffusers and SfBC) to showcase its versatility.** We maintained the original critic training pipelines for diffusers and SfBC and the behavior training pipeline for SfBC, then derived a new inference policy using our method. Experimental results are shown below. We test all tasks in 3 random seeds using a single shared behavior model due to a severe time limit. We will update the complete results in the paper before camera-ready submission.
>
>
> | Environment      | SfBC  | **SfBC+SRPO** (new)| Diffuser | **Diffuser+SRPO** (new) | IDQL  | **IDQL+SRPO**(original paper) |
> |------------------|-------|-----------|----------|---------------|-------|-----------|
> | HalfCheetah-ME   | 92.6  | 93.3      | 79.8     | 92.1          | 95.9  | 92.2      |
> | Hopper-ME        | 108.6 | 95.7      | 107.2    | 91.2          | 108.6 | 100.1     |
> | Walker2d-ME      | 109.8 | 110.3     | 108.4    | 112.8         | 112.7 | 114.0     |
> | HalfCheetah-M    | 45.9  | 60.9      | 44.2     | 58.0          | 51.0  | 60.4      |
> | Hopper-M         | 57.1  | 89.3      | 58.5     | 57.6          | 65.4  | 95.5      |
> | Walker2d-M       | 77.9  | 83.3      | 79.7     | 75.4          | 82.5  | 84.4      |
> | HalfCheetah-MR   | 37.1  | 46.4      | 42.2     | 45.9          | 45.9  | 51.4      |
> | Hopper-MR        | 86.2  | 100.0     | 101.3    | 93.7          | 92.1  | 101.2     |
> | Walker2d-MR      | 65.1  | 80.3      | 61.2     | 82.9          | 85.1  | 84.6      |
> | **Average**          | 75.6  | 84.4      | 75.3     | 78.8          | 82.1  | 87.1      |
> | **Inference time**          | 315.0  | 1.0      | 1669.0     | 1.0          | 47.5  | 1.0      |
>
>
> **Q2: The policy extraction process ... does not provide an adequate contribution to the community.**
>
> **A2** We would like to share some of the observations that drive us to study computational efficiency in diffusion-based RL.
>
> 1. When replicating experiments in IDQL, we find that policy evaluation takes 8 hours if we evaluate 50*10 episodes in total in an experiment, while policy training only takes 6 hours.
>
> 2. Numerous diffusion-RL works [3,4,5,6,7] describe acceleration techniques in their paper such as parallel computing to accelerate training (details in section 5.2 of the paper).
>
> 3. Recently, a series of methods have been proposed, similarly aiming to solve the computational efficiency problem in diffusion-based RL. For example, [7] proposes an approximate sampling approach. [8,9] use consistency models to replace diffusion models.
>
> To conclude, we think **computational efficiency, especially action sampling speed which we increase by more than 25x times, is a big concern for diffusion-based RL community.**
>
>
> [3] Planning with diffusion for flexible behavior synthesis (ICML 2022)
>
> [4] Offline reinforcement learning via high-fidelity generative behavior modeling  (ICLR 2023)
>
> [5] Contrastive energy prediction for exact energy-guided diffusion sampling in offline reinforcement learning. (ICML 2023)
>
> [6] Diffusion policies as an expressive policy class for offline reinforcement learning. (ICLR 2023)
>
> [7] Efficient diffusion policies for offline reinforcement learning (NeurIPS 2023)
>
> [8] Consistency Models as a Rich and Efficient Policy Class for Reinforcement Learning (https://arxiv.org/abs/2309.16984)
>
> [9] Boosting Continuous Control with Consistency Policy (https://arxiv.org/abs/2310.06343)

---

> > ### Comment · Reviewer_cVKT · 2023-11-17
> >
> > I acknowledge the answers provided by the authors to clarify some of my confusions. The main contribution of this paper, I think, is to derive a novel policy optimization objective from a behavior policy modeled via a pretrained diffusion model, and the objective is designed to increase the inference speed of diffusion-based polices. This is a critical issue in diffusion-based policies, and the paper has already mentioned and compared some work with similar motivation. One question is that why the work [7] mentioned in Section 5.2 is not compared in the experiments? Are there some particular reasons for the choice of baselines?

---

> ### Author Response · Authors · 2023-11-16
> **Official Response to Reviewer cVKT (2/2)**
>
> **Q3: There is no experiment concerning the robot scenarios especially with real data (to demonstrate the computational efficiency of the algorithm).**
>
> **A3:** (To avoid possible confusion, the D4RL tasks in our paper are basically all simulation-based tasks concerning the robotics scenarios. We hypothesize that the reviewer actually means experiments on real robots and real data instead of simulation data.)
>
> We definitely agree with the reviewer that experimental results on real robots could provide valuable insights.
> However, we currently lack the necessary hardware setups and expertise to conduct such experiments.
> Despite this limitation, we believe our evaluation remains robust for several reasons:
>
> 1. Among all the 10+ baselines we compare with, **only a single baseline method (Decision Diffuser[1]) has conducted experiments on real robots.**
> 2. In this paper, we focus on the theoretical derivation and benefits of the SRPO algorithm because it is theoretically novel. **Applying SRPO in real-world robotic hardware involves extensive engineering practices that are mostly orthogonal to the core focus of our original article.** These practices include hardware setup, data collection, and bridging the sim2real gap. Addressing these aspects **might divert the attention from the theoretical contributions of our paper**, which we aim to emphasize.
> 3. The primary aim of our work is to address the computational efficiency challenges in diffusion-RL methods. **This contribution is significant on its own, even without the deployment in real-world robotics.** One of the notable advantages is the reduction in evaluation time, which, in previous works like IDQL, constituted up to 50 $\%$ of the training time. This efficiency is also crucial for integrating diffusion models in online RL, where using a diffusion policy during the data collection phase is currently impractical due to time constraints[2].
> 4. Compared with other diffusion RL methods, **the benefit of fast inference speed is inherent to our algorithm design, regardless of whether our method is deployed in real-world robots.**
>
> Overall, while we recognize the value of real-world experiments, we sincerely hope that the reviewer can understand our dilemma here and finds it acceptable that we demonstrate effectiveness and efficiency only in simulation-based tasks. We believe this does not harm the key contribution of our method.
>
>
> [1] Is conditional generative modeling all you need for decision making? (ICLR 2023)
>
>
> [2] Policy Representation via Diffusion Probability Model for Reinforcement Learning (https://arxiv.org/pdf/2305.13122.pdf)
>
>
>
> **Q4: In Table 1, where do the results of baselines come from?**
>
> **A4** They all come from the original paper except for BEAR, which we refer to D4RL benchmark's white paper due to unavailability in the original paper.
>
>
> **Q5: It is considered that the pretrained diffusion model for behavior policy still requires iterative sampling during the pretraining phase which is visualized in Fig 5 when compared to IDQL.**
>
> **A5:** Neither IDQL nor our method requires iterative sampling during the pretraining phase.
>
> Reference code for IDQL:  https://github.com/philippe-eecs/IDQL/blob/6b071e7a1eea57bd93956f2b0a9e07a68081a096/jaxrl5/agents/ddpm_iql/ddpm_iql_learner.py#L426
>
> The pipeline (simplified) is
>
> 1. Sample $s, a_0$ from dataset, $t$ from $(0,1)$, Gaussian noise $\epsilon$
>
> 2. Add noise to $a\_0$ based on $t$ and $\epsilon$ to form $a\_t$
>
> 3. Optimize MSE loss between $\epsilon$ and $\epsilon\_\theta(a\_t)$
>
> **Q6: In Fig 5, are the results averaged across all locomotion tasks?**
>
> **A6** Yes, they are.
>
> Note that for hopper and walker2d environments, the episode may terminate in less than 1000 steps during evaluation (e.g., the robot falls to the ground). This causes too much randomness for evaluation, so we manually normalize the evaluation time hypothesizing that all episodes have a length of 1000.

---

> ### Author Response · Authors · 2023-11-17
> **Thank you for your engagement! Response to further questions.**
>
> We are glad to see we can clarify some confusion of the reviewer and that the contribution of our paper is acknowledged.
>
> **If we have resolved the reviewer's concern, we sincerely hope that the reviewer could consider raising the score (or increasing confidence).**
>
> **Q7: Why is work [7] not compared with?**
>
> **A7:** Work [7] is a very interesting work also studying the computational efficiency of diffusion RL. However, it was not included in our comparisons due to specific reasons:
>
> 1. Work [7] mainly aims to improve computational efficiency during the **training phase**, with basically no improvement in evaluation speed (Our method mainly increases evaluation speed).   As quoted in Section 5.1 of [7]: *EDP
> w/o DPM and DQL (JAX) are on par with each other regarding sampling speed*.  The official code, linked below, demonstrates that the proposed acceleration techniques are not utilized during evaluation. Consequently, EDP's inference speed matches that of D-QL, which we have already compared in Figure 1, achieving a 25-fold speed increase.
>
> https://github.com/sail-sg/edp/blob/b019648e3db9f8d035d3bf0897a8643bc5ab2369/diffusion/trainer.py#L349
>
> https://github.com/sail-sg/edp/blob/b019648e3db9f8d035d3bf0897a8643bc5ab2369/diffusion/trainer.py#L100
>
> 2. Regarding the performance metrics in Table 1, Work [7] was omitted mainly due to page limitations. We prioritized other baselines that exhibited higher D4RL performance. Nonetheless, we include a comparison with Work [7] in the table below for reference:
>
>
> | Algorithm                   | Mujoco (average) | Antmaze (average) |
> |-----------------------------|------------------|-------------------|
> | TD3+BC + [7]                | 85.5             | 29.8              |
> | CRR + [7]                   | 78.3             | 43.8              |
> | IQL + [7]                   | 79.9             | 73.4              |
> | (D)IQL with SRPO (ours)     | 87.1             | 73.6              |
>
>
> 3. Another minor consideration is the codebase of Work [7], which is based on JAX, known for faster execution compared to the Pytorch code used in our selected baselines.
>
>
> **Q8: Are there some particular reasons for the choice of baselines?**
>
> **A8:** Our selection of baselines was guided by their relevance to our algorithm and the extent to which they are recognized and understood in the field:
>
> 1. The five chosen diffusion-based RL methods are all well-acknowledged diffusion RL baselines.
>
> 2. As an early and influential paper, BEAR formally discusses behavior regularization policy optimization methods in offline RL, utilizing a VAE-based approach.
>
> 3. TD3+BC is a renowned Gaussian-based regularization method, widely recognized in the community.
>
> 4. IQL is a well-known weighted regression method and is frequently used as a baseline in recent studies.

---

> ### Comment · Reviewer_cVKT · 2023-11-20
>
> Thanks for the authors' explanations. I have raised my score accordingly.

---

> ### Author Response · Authors · 2023-11-21
> **Could you update the official review rating?**
>
> Dear reviewer,
>
> We are glad that our responses helped and would like to thank you for raising the score of our paper. However, we notice that the official OpenReview rating has not been updated, this will make the system and the AC unable to track the ratings for our paper.
>
> **Could you update the rating score of our paper to the OpenReview system, please?**
>
> 1. Navigate to the **original** official review section (the first one above).
>
> 2. Click on the *edit* button.
>
> 3. Select the new rating (8) to replace the original one (6).
>
> Thank you in advance!

---

### Official Review · Reviewer_fYZE · 2023-10-30

**Soundness:** 3 good
**Presentation:** 2 fair
**Contribution:** 2 fair
**Rating:** 3
**Confidence:** 3

**Summary:**

This paper proposes the use of pre-trained diffusion model for the behavior regularized offline policy optimization objective. The work shows a careful derivation of how pre-trained diffusion models, using the score function, can be replaced in the existing behavior regularized objective. The trick is to use the behavior policy distribution’s score function, and the paper claims to show faster compute time compared to existing approaches. Experimental results are demonstrated using the D4RL benchmark.

**Strengths:**

Algorithmically, the paper provides an interesting insight showing that in behavior regularized policy optimization objective, the gradient of the diverse term is indeed related to the score function of the behavior policy distribution. This therefore allows the use of pre-trained diffusion models to be used in these objectives.

The challenge of measuring the divergence term in offline regularized objective is generally difficult, where typically a separate model is needed to approximate the behavior model. Equation (9) provides a simple trick following equation (8) to show exactly where pre-trained diffusion models can be used for the behavior distribution, where the diffusion model e(a|s, t) can approximate the grad log term of the behavior policy. This is an interesting and novel insight to derive the algorithm, making use of the widely available pre-trained diffusion models these days.

**Weaknesses:**

The paper is a bit hard to follow; while the claims are justified, the paper is not so well written and seems convoluted. I believe this is also because the key idea/trick of the paper is to use pre-trained diffusion models in existing offline rl objectives, so the paper tries to lay out the context for that. However, it makes the paper rather difficult to follow, to completely understand the full contribution of the work.


Since the key idea is to use existing pre-trained diffusion models, I expected that other than the algorithmic contribution, the paper can do a much thorough job at more experimental evaluations? It would be useful to see existing setups where behavior regularized policy optimization is used, including toy examples, and perhaps provide a comprehensive qualitative study of the use of different pre-trained diffusion models in this context?

Experimental results probably need to be more thorough; there are only marginal benefits from the SRPO objective and it is not clear whether the proposed approach leads to empirical benefits. It would be helpful to do more qualitative studies on the objective and the use of different pre-trained diffusion models.

**Questions:**

See questions in the weakness section.

The key algorithmic pipeline is to integrate the SRPO technique with implicit Q learning. I wonder what happens if the SRPO objective is used in other behavior regularized offline rl objectives? Since the key algorithmic novelty comes through the derivation, it would be helpful if the authors can do more thorough experimental evaluation. Otherwise, the novel contributions of the paper are unclear.

---

> ### Author Response · Authors · 2023-11-16
> **Official Response to Reviewer fYZE (1/2)**
>
> We thank reviewer fYZE for his valuable feedback and his acknowledgment of our theoretical contribution. Although the current rating does not align with our expectations, we find that concerns primarily revolve around experimentation and presentation. Constructive suggestions have been provided. We seek to address the reviewer's concern by expanding experimental evaluations and improving presentation.
>
> **Q1: While the claims are justified, the paper is not so well written and seems convoluted.  ..the key idea/trick of the paper is ..., so the paper tries to lay out the context for that. However, it makes the paper rather difficult to follow**
>
> **A1:** We sincerely thank the reviewer for this valuable suggestion. Recognizing the concerns you have raised about the clarity and organization of our paper, we have undertaken significant revisions, **particularly in Section 3**, to enhance readability and coherence.
>
> Previously, we structured the paper to evolve naturally from our dual objectives: integrating diffusion models and pursuing computational efficiency, which then led to the development of our SRPO algorithm. However, we acknowledge that this structure, rather than clarifying, inadvertently made the paper convoluted.
>
> In response to your feedback, we have implemented the following revisions:
>
> 1. We have shifted the narrative to **directly introduce the SRPO algorithm**, followed by an analysis of its benefits and a comparison with prior methodologies. This "straight to the point" approach clarifies our contributions without the need for extensive background exposition.
> 2. We have removed unnecessary background information and considerations that were not directly relevant to the derivation of the SRPO algorithm.
> 3. Sections 3.1 and 3.2 have been merged into a single section to make our presentation more cohesive and compact.
>
> **Q2: It would be useful to see existing setups where behavior regularized policy optimization is used, including toy examples**
>
> **A2:** Based on the reviewer's suggestion, **we have conducted additional experiments. Visualization results are provided** in Appendix A of the paper.
>
> 1. We compare our proposed method against other behavior regularization methods, including BCQ, BEAR, and TD3+BC, across six tasks set in 2D bandit environments (Figure 4, Figure 9).
>
> 2. To illustrate the advantages of using diffusion models over VAE models for behavior modeling, we conducted qualitative experiments. The diffusion and VAE models tested were nearly equivalent in terms of the number of parameters and network depth (Figure 9).
>
> Our findings highlight that Gaussian-based regularization methods, such as TD3+BC, tend to underperform in scenarios with complex behavior distributions, even in 2D settings. Rather than adhering closely to the behavioral actions, the learned policy often gravitates towards **the center of mass** of behavioral actions. VAE-based regularization methods, on the other hand, more frequently result in policies that select actions outside the dataset support, compared with our method. We attribute this to the generative capacity gap between diffusion and VAE models.
>
> **Q3: It would be helpful to do more qualitative studies on the objective.**
>
> **A3:** The two most important design choices in our policy optimization objective are the weighting function $w(t)$ and the temperature $\alpha$. **We provide more qualitative results in the paper.**
>
> 1. On the weighting function $w(t)$. In order to illustrate the empirical benefits of ensembling multiple diffusion times for score regularization, we calculate and visualize the density map learned by diffusion behavior models at different diffusion times $t$. Then we provide qualitative results to compare the regularization effect without the ensembling technique proposed in Section 4.2 of the paper. **(Figure 5, Figure 10, and Figure 11)**
>
> 2. On the temperature $\alpha$. We vary the temperature $\alpha$. We provide qualitative results in Figure 12 and quantitative results in **Figure 15**.

---

> ### Author Response · Authors · 2023-11-16
> **Official Response to Reviewer fYZE (2/2)**
>
> **Q4: More thorough experimental results ... on the use of different pre-trained diffusion models.**
>
> **A4:** (We interpret the reviewer's reference to different pre-trained diffusion models as applying our method to various previous diffusion-based RL literature.)
>
> To further validate the effectiveness of our policy extraction scheme, we have **additionally applied our proposed score-regularization method to two additional diffusion-based Offline RL methods beyond IDQL**. Experimental results are given below.
>
> These baselines are recognized diffusion-RL methods. Previously, they rely on an iterative diffusion sampling scheme for action selection, a process that is both time-consuming and highly stochastic. By integrating our proposed policy extraction scheme, **we observe improvements in both overall performance and inference speed**.
>
>
> | Environment      | SfBC  | **SfBC+SRPO** (new)| Diffuser | **Diffuser+SRPO** (new) | IDQL  | **IDQL+SRPO**(original paper) |
> |------------------|-------|-----------|----------|---------------|-------|-----------|
> | HalfCheetah-ME   | 92.6  | 93.3      | 79.8     | 92.1          | 95.9  | 92.2      |
> | Hopper-ME        | 108.6 | 95.7      | 107.2    | 91.2          | 108.6 | 100.1     |
> | Walker2d-ME      | 109.8 | 110.3     | 108.4    | 112.8         | 112.7 | 114.0     |
> | HalfCheetah-M    | 45.9  | 60.9      | 44.2     | 58.0          | 51.0  | 60.4      |
> | Hopper-M         | 57.1  | 89.3      | 58.5     | 57.6          | 65.4  | 95.5      |
> | Walker2d-M       | 77.9  | 83.3      | 79.7     | 75.4          | 82.5  | 84.4      |
> | HalfCheetah-MR   | 37.1  | 46.4      | 42.2     | 45.9          | 45.9  | 51.4      |
> | Hopper-MR        | 86.2  | 100.0     | 101.3    | 93.7          | 92.1  | 101.2     |
> | Walker2d-MR      | 65.1  | 80.3      | 61.2     | 82.9          | 85.1  | 84.6      |
> | **Average**          | 75.6  | 84.4      | 75.3     | 78.8          | 82.1  | 87.1      |
> | **Inference time**          | 315.0  | 1.0      | 1669.0     | 1.0          | 47.5  | 1.0      |
>
> Baselines details:
>
> 1. Diffusers trains a discrete-time diffusion behavior model and has a transformer architecture. The critic model is trained by following the existing classifier guidance method.
>
> 2. SfBC also trains a continuous-time diffusion behavior model with an encoder-decoder architecture. The critic model is training through implicit planning.
>
> Due to time constraints, we tested all tasks using three random seeds and a single shared behavior model. We intend to provide complete results before the camera-ready submission.
>
>
> **Q5: There are only marginal benefits from the SRPO  objective and it is not clear whether the proposed approach leads to empirical benefits.**
>
> **A5:** We respectfully disagree with the reviewer on this comment. We contend that the SRPO objective offers clear and significant benefits over previous approaches:
>
> 1. A pivotal advantage of the SRPO objective is its **substantially faster inference speed** – at least 25 times quicker than various leading diffusion-based RL methods. This is not a marginal benefit but a fundamental improvement. **This benefit is inherent to our design;** unlike previous diffusion-based RL methods, which require iterative diffusion sampling during training or evaluation, our approach streamlines this process.
>
> 2. **The experimental results, particularly after this revision, support our claims**. Our experiments detailed in responses A3 and A4, as well as in Table 1 and Appendix A, consistently demonstrate that SRPO can lead to empirical performance improvements in both D4RL benchmarks and 2D environments, outperforming previous methods.
>
> 3. Compared to traditional VAE-based behavior regularization methods, our carefully derived score regularization method enables the effective utilization of more powerful diffusion modeling techniques. To the best of our knowledge, **similar objectives have not been explored in prior Offline RL literature**.

---

> ### Author Response · Authors · 2023-11-20
> **Please let us know if we have resolved your concerns**
>
> Dear reviewer,
>
> We were wondering if our responses and revision have resolved your concerns since only three days are left for the discussion phase. Based on your suggestions, we have undertaken significant revisions to our manuscript. We conduct three additional types of experiments (both qualitatively and quantitatively) and enhance the way we organize our narrative structure. Please let us know if these changes meet your expectations. We are eager to engage in further discussions and continue improving our work.
>
> Best regards,
>
> The Authors

---

> ### Author Response · Authors · 2023-11-21
> **Your feedback is critical to us**
>
> Dear reviewer,
>
> We were wondering if our responses and revision have resolved your concerns since only two days are left for discussion. we have undertaken significant revisions to our manuscript based on your suggestions. We conduct three additional types of experiments (both qualitatively and quantitatively) and enhance the way we organize our narrative structure. Please let us know if these changes meet your expectations. We are eager to engage in further discussions and continue improving our work.
>
> Best regards,
>
> The Authors

---

> ### Author Response · Authors · 2023-11-22
> **Awaiting your valuable feedback before deadline**
>
> Dear Reviewer,
>
> We fully appreciate the demands on your time, particularly during this busy period. As today marks the **final day of the discussion stage**, we are writing to kindly request your feedback on our responses. Your insights are crucial to addressing any remaining concerns you may have regarding our submission.
>
> Please note that after November 22nd, while you are welcome to reassess or comment on our work at your convenience, our ability to respond may be limited. Therefore, we are particularly keen to engage in a constructive dialogue with you before the imminent deadline. Your feedback is not only valuable to us but also essential for the final evaluation of our work.
>
> Best,
>
> Submission 6831 Authors

---

### Official Review · Reviewer_bNEQ · 2023-11-01

**Soundness:** 4 excellent
**Presentation:** 4 excellent
**Contribution:** 4 excellent
**Rating:** 8
**Confidence:** 3

**Summary:**

This paper introduces a score regularized policy optimization algorithm (SRPO) based on behavior diffusion, aiming to address the problem of heterogeneous behavior distributions in offline reinforcement learning. The algorithm utilizes a pre-trained behavior diffusion model to score-normalize policy gradients and presents a practical method based on behavior diffusion and implicit Q-learning. The paper provides a detailed explanation of SRPO's principles, implementation, and optimization techniques, and it validates the effectiveness of the algorithm through experiments.

**Strengths:**

In the offline reinforcement learning setting, the paper introduces a novel approach that leverages the diffusion model. This method uses the powerful modeling capabilities of the diffusion model while avoiding the extensive time-consuming iterative inference stage.

**Weaknesses:**

The final policy used by the algorithm is still based on a Gaussian distribution. This Gaussian policy might not capture complex distributions as effectively as the diffusion model when dealing with complex offline datasets. The key concern here is whether the complex distribution information modeled by the pretrained diffusion behavior can be adequately captured by a policy based on a Gaussian distribution.

**Questions:**

1. DIQL should be IDQL in the first and second paragraph of section 6.1.
2. Why do different Gaussian-based policies have significantly varying inference times, as shown in Figure 1?

---

> ### Author Response · Authors · 2023-11-16
> **Official Response to Reviewer bNEQ**
>
> We thank reviewer bNEQ for his expertise and insightful comments. We give a point-to-point response to the raised concerns below and hope they may help increase the reviewer's confidence in our work.
>
> **Q1: This Gaussian policy might not capture complex distributions as effectively as the diffusion model when dealing with complex offline datasets.**
>
> **A1:**
> We partially agree with the reviewer. In our opinion, diffusion models are beneficial for offline RL for mainly two reasons:
>
> 1. The behavior dataset $\mu(a|s)$ is complex.
>
> 2. The optimal policy $\pi^* \propto  \mu(a|s) \ e^{\beta Q(s, a)}$ is potentially complex (multi-modal).
>
> If referring to problem 1, our proposed method captures complex behavior distributions effectively and **efficiently** by utilizing a diffusion behavior model. No Gaussian model is used.
>
> If referring to problem 2, we use a Dirac (gaussian) policy to capture the mode of $\pi^*$:
>
> $\pi^{\text{gauss}}\_{\theta} (s) = \arg \max\_a \mu(a|s) \ e^{\beta Q(s, a)} $ instead of $\pi_\theta^{\text{diff}}(a|s) \propto \mu(a|s) \ e^{\beta Q(s, a)}$.
>
> **We agree with the reviewer that the Gaussian policy cannot capture complex $\pi^{\*}$. This is simply not our intention.**
> If there are two optimal policies for one single task, our method will simply choose either one of them deterministically. Our observation is that in many scenarios, inference speed is of much higher priority than policy diversity. For instance, in robotics, we often do not care about how many ways our robot can solve a task. Contrastively, we do care about whether our policy is optimal and efficient (hardware-friendly). Also, current evaluation metrics of the D4RL benchmark only emphasize policy optimality and ignore policy diversity.
>
> Still, we believe a better trade-off between inference speed and policy diversity exists and could be a future direction of our work.
>
> **Q2: The key concern here is whether the complex distribution information modeled by the pretrained diffusion behavior can be adequately captured by a policy based on a Gaussian distribution.**
>
> **A2:** We really appreciate this insightful comment.
>
> Theoretically, since SRPO can be thought of as performing gradient descent in the action space: $\arg \max\_a [\log \mu(a|s) + \beta Q(s, a)]$, it is possible that the learned policy gets stuck in a local optimal and thus fails to capture the mode of $\pi^*$.
>
> To address this problem, our work proposes to ensemble diffusion score estimators at diffusion times $t\in (0,1)$ to make the gradient landscape smoother. This part is explained in section 4.2 and analyzed in Appendix B of the paper. **We have newly added an illustration in Figure 5 and Figure 10 in the revisited paper.** Ablation studies in Section 6.3 also support the effectiveness of our proposed method.
>
> **Empirically, we feel confident about SRPO's ability to capture complex distribution information modeled by the pre-trained diffusion in offline RL.** Besides reported evaluation numbers, our confidence also comes from DreamFusion (a text-to-3D research winning the Outstanding Paper Award of ICLR2023), which inspires our work. DreamFusion seeks to learn a Dirac 3D model from a pre-trained large-scale image diffusion model. Their method similarly adopts a score distillation method to supervise the training of another Dirac (Gaussian) 3D model. Experimental results show that Gaussian-based policy can already capture complex and high dimensional distribution successfully **even in complex image space** (https://dreamfusion3d.github.io/).
>
>
> **Q3: DIQL should be IDQL in the first and second paragraph of section 6.1.**
>
> **A3:** We sincerely thank the reviewer for the careful reading. The typo is fixed now.
>
> **Q4: Why do different Gaussian-based policies have significantly varying inference times, as shown in Figure 1?**
>
> **A4:** Actually inference times for Gaussian-based policies are very similar. They only look that way because the x-axis is in the log scale in Figure 1.
>
> |                | Ours | BEAR | IQL  | CQL  | AWAC | BCQ | DT   | IDQL  | Diffuser |
> |:---------------|:-----|:-----|:-----|:-----|:-----|:----|:-----|:------|:---------|
> | Inference time | 1.28 | 1.28 | 1.43 | 2.32 | 2.86 | 9.0 | 31.6 | 60.83 | 2136.4   |
>
> ps. AWAC takes a longer time for action sampling simply because it is a 4-layer MLP, while ours is a 2-layer MLP.

---

> ### Author Response · Authors · 2023-11-20
> **Please let us know if we have resolved your concerns**
>
> Dear reviewer,
>
> We were wondering if our responses have resolved your concerns. We hope that these clarifications can serve to enhance your confidence in the ratings of our work. Should you have any further questions or require additional insights, please feel free to reach out. We are fully committed to providing any necessary information to support your assessment.
>
> Warm regards,
>
> The Authors

---

### Author Response · Authors · 2023-11-17
**Revision Summary**

We would like to thank all the reviewers for their detailed comments and valuable suggestions.

We have made a number of changes to address reviewers' suggestions and concerns. A short summary of the modifications made:

1. We have rigorously revised the manuscript to make the narrative more straightforward and less convoluted. This revision particularly emphasizes our high-level assumptions and motivations, ensuring that these elements are highlighted and comprehensible (particularly Section 3).
2. We conduct qualitative experiments to better illustrate the benefits of ensembling various diffusion times (Figure 5,10,11).
3. We conduct additional experiments by applying our proposed score-regularization method to two additional diffusion-based Offline RL methods beyond IDQL originally Table 1. We quantitatively compare their performance and inference speed across 9 D4RL tasks.
4. We conduct additional experiments to compare our method with 3 previous approaches by visualizing their performance across 6 tasks in 2D settings. (Figure 4, 9).

We look forward to further discussions with the reviewers!

---

### Meta-Review · Area_Chair_B7X9 · 2023-12-06

**Metareview:**

The authors propose to use diffusion models for policy regularization in the setting of offline RL, effectively harnessing the expressive capacity of the diffusion model while avoiding the computational expense of the iterative sampling procedure. The reviewers generally found the authors’ approach interesting and clear, however, they also cited concerns about the scope of the empirical evaluation and novelty of the approach. The authors have largely addressed these concerns in the rebuttal phase.

**Justification For Why Not Higher Score:**

The paper presents a useful approach, however, the main benefit is in improved computational efficiency by combining previous methods in improved ways. While this is an important contribution, I see this as somewhat lacking in the novelty required for a spotlight or oral.

**Justification For Why Not Lower Score:**

The authors have presented a method for using diffusion-based policies while improving the computational efficiency of the overall policy. This is a useful contribution, which the reviewers generally found to be warranting publication.

---

### Decision · Program_Chairs · 2024-01-16

Accept (poster)